# Functional Alignment Can Mislead: Examining Model Stitching

Damian Smith [1]  Harvey Mannering [1]  Antonia Marcu [1]

## Abstract

A common belief in the representational comparison literature is that if two representations can be functionally aligned, they must capture similar information. In this paper we focus on model stitching and show that models can be functionally aligned, but represent very different information. Firstly, we show that discriminative models with very different biases can be stitched together. We then show that models trained to solve entirely different tasks on different data modalities, and even representations in the form of clustered random noise, can be successfully stitched into MNIST or ImageNet-trained models. We proceed by showing that alignments can also be found in the case of autoencoders where the encoder and decoder are trained on different tasks. We end with a discussion of the wider impact of our results on the community's current beliefs. Overall, our paper draws attention to the need to correctly interpret the results of such functional similarity measures and highlights the need for approaches that capture informational similarity.

## 1. Motivation

Measuring representational similarity is a complex task and there is no consensus on how it should be done (Sucholutsky et al., 2023). More interestingly, there seems to be no consensus on **when** two representations can be considered similar. In this paper we *do not* consider two representations to be similar if they *do not* capture the same information. For example, if one model learns to classify birds by their song and another learns to classify them by their visual appearance, we consider those models to be different. Equally, if two models learn to visually classify birds but one of them uses the birds' shape to make predictions, while the other uses plumage, we would consider those models to be

---

[1]Vision, Learning, and Control (VLC) Research Group, University of Southampton. Correspondence to: Antonia Marcu <a.marcu@soton.ac.uk>.

*Proceedings of the 42nd International Conference on Machine Learning*, Vancouver, Canada. PMLR 267, 2025. Copyright 2025 by the author(s).

*Do networks that learn to solve all these tasks really represent the same information?*

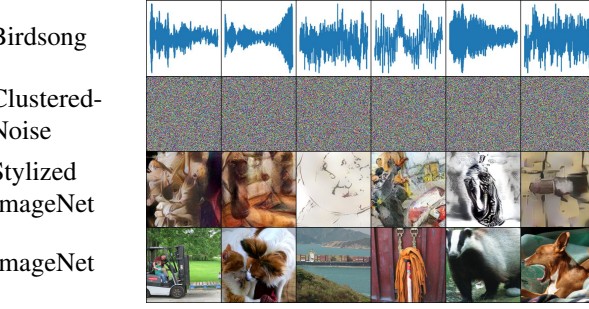

Birdsong
Clustered-Noise
Stylized ImageNet
ImageNet

*Figure 1.* Examples of inputs from 4 different tasks we consider. We show that models trained on the first three tasks can be stitched together with models trained on ImageNet despite depicting very different input-level information.

different, since they also use different input patterns for their predictions. Throughout the paper we refer to input-level patterns as "semantic" information.

We argue that without capturing semantic similarity, some of the existing model comparison measures can end up considering a network that successfully distinguishes birds based on their songs to be equivalent to one that successfully distinguishes ImageNet images. In particular, we focus on model stitching but we argue in our discussion section that the same criticism might be extended to other measures and hypotheses as well.

As an alternative to established model comparison methods such as CKA (Kornblith et al., 2019), *functional* model comparison is gaining increasing attention (Bansal et al., 2021; Hernandez et al., 2022; Klabunde et al., 2025; Csiszárik et al., 2021). In this paper we focus on the functional perspective as an emerging direction and will therefore not cover CKA, SVCCA (Raghu et al., 2017), or similar methods. Broadly speaking, the functional perspective argues that two intermediate layers should be considered similar if they **lead to** matching outputs. While this perspective seems useful, especially in a classification setting, we argue that there is one fundamental problem with deep learning models which raises questions about its utility: shortcut learning (Geirhos et al., 2020).

Shortcut learning is an umbrella term for learning decision

rules that rely on spurious correlations, causing the model to perform poorly outside of the setting it was trained on. Geirhos et al. (2020) argue that many challenges in deep learning boil down to shortcut learning. Moreover, the simplicity bias literature argues that SGD-trained models have a tendency to learn simpler rules (e.g. Valle-Perez et al., 2018; De Palma et al., 2019) (often shortcuts) over more complex rules that capture some of the true, salient, or intended information. Therefore, there is an increasing consensus that **models are prone to picking up shortcuts in the data**, whether we are aware of their existence or not. As a result, it is important to be able to distinguish between models that learn different patterns because some could fail to generalise in the wild.

In this paper we look at model comparison starting from the understanding that the models we compare have a propensity for learning shortcuts. In particular, we focus on model stitching (Lenc & Vedaldi, 2015), which is an increasingly popular method for comparing models' functional similarity. Informally, model stitching compares models $A$ and $B$ by "stitching" together the first part of model $A$ (the *sender*) with the last part of model $B$ (the *receiver*) using a linear transformation referred to as a "stitch". The stitch is trained such that it learns a mapping between representations of models $A$ and $B$. Model $A$ is considered similar to model $B$ if their stitched combination achieves an accuracy comparable to that of model $B$. In other words, model stitching assesses how compatible two models are from a functional perspective by comparing how the function represented by their stitched combination behaves compared to the $B$ function when evaluated at the same input points. Functional compatibility is often taken to indicate that models capture similar information (e.g. Lenc & Vedaldi, 2015; Bansal et al., 2021; Hernandez et al., 2022).

In this paper we design experiments which demonstrate that the stitching **compatibility** (or alignment) of models **is inappropriate** for evaluating the extent to which two networks capture similar information. Importantly, a model that learns a spurious correlation can appear to be fully compatible with one that learns the intended information. Given the prevalence of shortcut learning, we argue that a meaningful model comparison should distinguish between models that capture different input patterns.

We first show that models that learn to use different information can easily be stitched together. We start with a toy problem where we can bias models towards learning to use either colour or shape information. We show that the models are considered similar from a stitching perspective. We then consider a more challenging setting where we ensure that the sender uses different information from the receiver, and also that its representations do not contain information relevant for the original receiver network. We do this by

removing all shape information from the input data to be represented, leaving only colour information. We then stitch into a model that was trained on shape information alone and that is unable to make correct predictions based on colour information. We achieve full stitching compatibility in this case as well. We then show that we can successfully stitch clustered, random noise into the models we train. This, therefore, shows that we can easily construct cases where the models' stitching compatibility does not reflect their informational similarity.

In the second part of the paper we demonstrate that the problem is not restricted to simple settings by performing similar experiments with more complex, "real-world" data. For example, we show that we can stitch a model trained on ImageNet (Russakovsky et al., 2015) to one trained to recognise bird songs.

Our contributions are:

- We argue that accounting for the extracted input patterns is of paramount importance when comparing representations, and show that model stitching does not achieve this;

- We show that we can construct problems where stitching cannot distinguish between models known to have learned different shortcuts (i.e. they use different information) (Section 4.1);

- We show that representations that depict entirely different information or even clustered, random noise can be stitched into a trained model, raising further questions about the usefulness of stitching as a measure of model similarity (Section 4.2);

- We extend our experiments to more complex, real-world, and benchmark datasets, considering distinct modalities and different tasks for both the discriminative and the generative case (Section 5);

- And, we end with a discussion of the wider implications of our results on functional alignment.

The code for our experiments is available at `https://github.com/DHLSmith/stitching.git`.

## 2. Model Stitching

Model stitching has seen many variants (e.g. Bansal et al., 2021; Csiszárik et al., 2021; Hernandez et al., 2022) since it was first proposed (Lenc & Vedaldi, 2015). For simplicity, in this paper we only consider stitching between identical architectures and at matching points in the network. Let $A_{\leq l}$ denote the composition of all layers in the trained network $A$ up to and including layer $l$. Following Bansal et al. (2021),

we choose the stitching layer $s$ to be a randomly initialised $1 \times 1$ convolutional layer preceded and followed by batch normalisation (Ioffe & Szegedy, 2015). Bansal et al. (2021) use a convolutional layer as it has restricted expressivity. The untrained stitched model is therefore given by $B_{>l} \circ s \circ A_{\leq l}$. We then train the stitched model by freezing the parts taken from the models $A$ (the sender) and $B$ (the receiver) and only optimising the stitching layer $s$. We then report the performance of the stitched model on the test data. If the test performance of the stitched model is greater or equal to that of the receiver (which becomes the baseline), it is considered that models $A$ and $B$ are *compatible* at layer $l$.

In this paper we focus on ResNet-18 and ResNet-50 (He et al., 2016) models, but include results for VGG19 (Simonyan & Zisserman, 2015) in Appendix D.1. Note that we stitch ResNet models before residual blocks and before the linear classifier. Therefore we refer to stitching before the first residual block as stitching "Res Block 1"; before the second residual block as "Res Block 2"; and stitching before the classifier as "Linear".

**Current stitching interpretation.** This work is inspired by Hernandez et al. (2022) who find that stitching can reach high accuracy even when stitching from later layers in Model A to much earlier layers in a Model B. They believe that this could either be because the common intuition about how models process input is wrong, or because model stitching is able to match representations which are "different from what is expected". Nonetheless, they conclude that functional similarity provides a meaningful way of comparing models. Their intuition is that "if two representations can be used for similar purposes then in some sense they encode similar information" (Hernandez et al., 2022). This agrees with Bansal et al. (2021) who argue "two networks with identical architectures, but very different internal representations, would fail to be stitching connected".

We argue that no meaningful conclusion can be drawn from analysing the stitch connectivity. When stitching is **not** successful, this could simply be because a good enough mapping between the representations was not found (Csiszárik et al., 2021). We claim that when stitching **is** successful, one cannot conclude that this is necessarily because the representations capture the same information. The next section shows the latter empirically.

## 3. Inducing a Learning Bias Through the Data

We choose to compare models on a typical shortcut learning problem: a variant of colour MNIST (Bahng et al., 2020). In essence, colour MNIST adds colour as an additional correlation between the input variable and the target variable. We choose to add colour as a fully-correlated background. For example, all instances of digit "0" have a red background,

all instances of "1" have a green background, etc. We refer to this dataset as "Correlated".

We use this problem to show that stitching is unable to distinguish between models that use different features to make the prediction. To do so, we want to simulate a scenario where **training various models leads to each learning a different pattern**. One possible way of biasing the model towards picking up different rules is to modify the architecture. However, modifying the architecture means that we can't perform a one-to-one model comparison. For this reason, we choose to modify the training data instead as a way of biasing the models. We therefore create different variants of the dataset that lead to the model classifying based on either colour information alone, shape information alone, or different levels of relying on the combination between colour and shape.

Importantly, note that we modify the data in such a way that each model would still perform well (over 98% test accuracy) on the original Correlated dataset. This means that the models could have been trained on the original Correlated data if we had an appropriate way of biasing them in a controlled way through the training procedure alone. Below we provide a brief description of the dataset versions created (see Figure 2a for visual examples). For full details on the creation of the datasets, see Appendix A.

**Colour MNIST (Correlated):** The colour of the background and class of the digit are correlated.

**Digit with Uncorrelated Colour (Digit):** Images containing random combinations of background colour and digit. Target is given by the digit's class. Note that the colours were chosen from the same set of colours used to generate the Correlated dataset.

**Colour with Uncorrelated Digits (Colour):** Images containing random combinations of background colour and digit. Target is given by the colour's class. Importantly, the model could learn to represent shape information, but this cannot help the model classify.

**Colour with no Digit (Colour-Only):** Images containing background colour and no digit. Target is the colour's class. The model cannot learn to base its predictions on shape information.

For clarity, note that the short version of the dataset's name (e.g. Digit) refers to the part of the input that is correlated with the label. We refer to the models by the name of the dataset they were trained on. Unless otherwise stated, the results are reported on ResNet-18 models. For full experimental details see Appendix B.

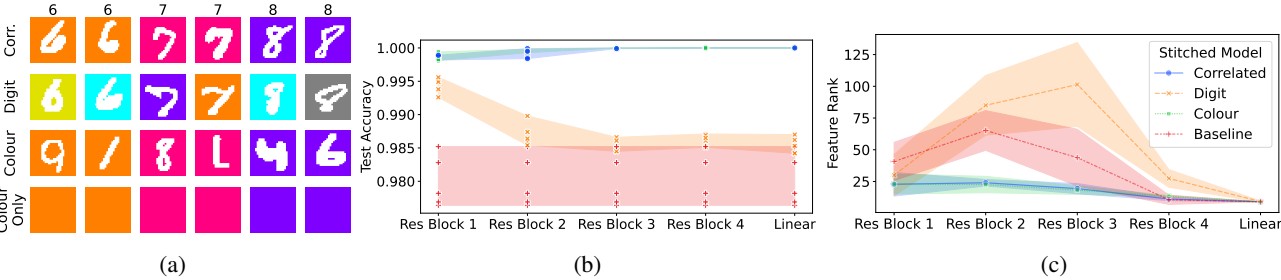

*Figure 2.* **(a)** Two example images for classes '6', '7', and '8' in each dataset type. **(b)** Stitching test accuracy when stitching at each of the 5 layers we consider, for all four types of model stitched into Digit model with Correlated data. Baseline is given by the test accuracy of the unstitched Digit model on the Correlated dataset, which is the task we assume to be given. Shaded area covers 100% of results range, markers show each result across multiple initialisations. We find that we can achieve stitch compatibility across different layers for all the models we consider. (For each stitched Digit model, accuracy remains above its own unstitched baseline) **(c)** Rank analysis of the feature maps output by the receiver's first layer upon stitching. Shaded areas mark 1 standard deviation.

## 4. Experiments

We start this section by adhering to Bansal et al. (2021)'s original setting. Namely, we assume we are given the task of classifying the Correlated dataset and we want to analyse the similarity of multiple models trained to solve this task. We show that stitching cannot distinguish between models capturing different biases in the data. To rule out the possibility that the results we observed are due to information leak, we then move away from the original stitching setting (Section 4.2). We remove from the input (and as a consequence, from the sender's representations) any information related to the receiver's task and manage to achieve stitching compatibility.

### 4.1. Stitching Cannot Distinguish Between Different Biases

In this section we want to verify if stitching can help distinguish between models that learn to classify using different information. We start by stitching various senders into the Digit model (i.e. the digit is correlated with the label and the background has a random colour). Note that for training and evaluating the stitch, we use the Correlated dataset as this is the data we assume we have access to, in accordance with the typical stitching procedure. All other datasets were created simply to train senders and receivers with different biases. We find that at all layers, all models we consider can easily be stitched into the Digit model (which becomes the receiver), obtaining a higher accuracy than that of the original Digit model (see Figure 2b). That is, even the Colour with Uncorrelated Digits model, which learned to classify based on colour information (ignoring shape), appears compatible with the model that learned to classify based on shape information only.

Taking this further, we train a model on patches of colour with no digit ('Colour-Only'). As opposed to Colour (which

includes an uncorrelated digit), the model can only learn to represent solid colour information. Nonetheless, when stitching on the original task, we obtain a test accuracy higher than that of the receiver alone, indicating stitching compatibility (see Figure C.1a).

**Numerical rank.** We analyse the rank of the feature maps after the first layer of the receiving network. This is a way of gauging the linear dependence of the sender's feature maps as seen through the lens of the receiver's representation. If two sender representations have similar rank when processed by the receiver network, we cannot necessarily claim that they are perceived in a similar way. We can only say that the receiver represents them with a similar level of linear dependence. However, if they do not have the same rank, we take that as additional evidence that the senders have learned different information **according to the receiver**. Details of the numerical rank estimation are provided in Appendix B.2.1.

In Figure 2c we show the rank computed when stitching at each of the 5 layers we consider. We find that for stitching before the $2^{nd}$ and $3^{rd}$ residual blocks there is a clear gap between the rank of the processed representations of Digit with Uncorrelated Colour and those of Correlated, for example. This indicates that when fed into the receiving network, the representations of Digit with Uncorrelated Colour have a different degree of linear dependence from those of Correlated and therefore are not perceived as equivalent by the receiver despite both being stitch-compatible.

The numerical rank of representations provides an estimation of their compression. Note that alternative estimators of compression could be considered. Each estimator, including numerical rank, comes with its limitations. Our objective is to include additional evidence that the various sender representations are not entirely equivalent when processed by the receiver. Following recent work (e.g. Masarczyk et al.,

2023; Feng et al., 2022), we choose to use numerical rank to illustrate the difference in compression.

Arguably, in the stitching cases considered so far colour information could still "leak" through a model that did not learn to use it for classification. The same argument holds for structured shape information potentially leaking. We next perform two follow-up experiments: with a receiver model trained only on shape information, we obtain full stitch compatibility on Colour-Only images (no digit is depicted, Figure 2a), despite no shape information being present in the data; and stitching clustered, random noise simulating a sender network's representation (see Section 4.2) into the receiving network. These experiments strengthen our findings that models can be easily stitched together even when they represent very different information.

### 4.2. Representations of Very Different Information Can Be Stitched Together

To check whether stitching between dissimilar models is due to the sender model "leaking" information expected by the receiver, we remove all information that could be expected by the receiver from the images we train and evaluate the stitch on. Concretely, we stitch Colour-Only models into receivers trained on Greyscale MNIST data but this time using Colour-Only data to train and test the stitch. There is no digit-like structure in the Colour-Only data which (if leaked) could be used by the MNIST receiver. The unstitched MNIST models have a baseline accuracy (when tested with MNIST data) of $99.0 \pm 0.1\%$; at every stitching layer, the Colour-Only sender increases this to 100% accuracy, resulting in stitching compatibility. In other words, despite the sender representations not containing any digit information at all (as the input does not contain that information in the first place) we are able to successfully stitch those representations into a Greyscale MNIST model.

Lastly, we stitch clustered, random noise (**Clustered-Noise**) into the receiver network. We create 10 random vectors to represent the mean of each of the 10 classes. The vectors have the same dimensionality as the representations that would normally be expected by the stitching layer. For each class we create 6K samples by adding random noise to the mean class vector, obtaining a dataset of the same size as the original Correlated dataset (see Appendix A). Note that we do not use Clustered-Noise to train a sender. Instead, Clustered-Noise simulates the representations of a sender network. We then stitch these simulated representations onto the Digit receiver. We can achieve **full stitching compatibility when using randomly generated representations**. As expected, the rank of these representations when fed into the receiver network is different from that of the learned representations (see Figure C.1b). We find similar results with VGG19 models (see Appendix D.1).

In light of these results, we argue that models that extract patterns from significantly different information can, in at least some cases, be easily stitched together. While it may be argued that the datasets chosen are too simple, it is known that SGD-trained models find simple rules or shortcuts (e.g. Valle-Perez et al., 2018; De Palma et al., 2019), and there is no algorithm to decide when stitching is applicable. Therefore, models' stitching compatibility should not be taken to mean that they learn similar rules or that they capture similar information.

Note that these last experiments (starting Section 4.2) deliberately move away from Bansal et al. (2021)'s original setting. The stitch is now trained and evaluated on a different task to the one that the receiver was trained on. This is necessary in order to rule out the effect of information leak. This extended setting allows us to emphasise that stitching alignment can abstract away from the input patterns that produced the sender's representations. Because of this, stitching cannot reliably distinguish between two models that capture different cues in the data. In the next section, we are going to explore this extended setting further by stitching together models that were trained to solve different tasks altogether.

## 5. Other Successful Stitchings: Discriminative Case

So far we focused on simple problems which allowed us to control what information is available to the models. The remainder of the paper focuses on "harder" problems. We would like to highlight, however, that in such cases no clear ground truth exists and constructing arguments based on what "intuitively makes sense" can be deceptive. We simply offer these as supporting evidence to our well-controlled experiments on which all our claims are based.

Note that Bansal et al. (2021) consider the performance of the receiver as the baseline. In our previous experiments, although the sender and the receiver were trained on different datasets they were still trained to be compatible with the same task, which is Correlated (both the digit class and the colour class are correlated with the label). As a reminder, our goal was simply to induce a strong bias towards solving the task using either the digit or the background colour. Naturally, when tested on fully correlated data, both models were successfully solving the task.

In a non-toy setting, when trying to stitch between models that are unlikely to represent the same input patterns, the receiver will inevitably perform poorly on the sender's task. This means that the baseline will usually be very weak. We choose to set a higher baseline, which is the performance of the receiver on its own task. For example, in the case when the ImageNet model is the receiver on Stylized data (which

*Table 1.* Results of stitching various models into and from ImageNet. We highlight in bold the representations that were stitch-compatible. Note that "Block $X$" and "Linear" means that we are stitching **before** the $X$th residual block and the classification layer respectively. Despite choosing a more challenging baseline than typically considered for stitching, we are still able to find some models stitch-compatible at various layers. *Accuracy results for this baseline are given by Stylized model's performance on ImageNet data which, although atypical, are higher than Stylized model's performance on the Stylized task (56.18% top-1 and 78.96% top-5 accuracy). Note that with the weaker baseline, we find compatibility at all layers.

| | | Block 1 | Block 2 | Block 3 | Block 4 | Block 5 | Linear | Baseline |
|---|---|---|---|---|---|---|---|---|
| ImageNet to | Top-1 | 58.39 | 58.61 | **61.53** | **65.31** | **70.43** | **74.34** | 60.18* |
| Stylized | Top-5 | 80.57 | 80.61 | **83.28** | **86.13** | **89.78** | **91.93** | 82.60* |
| Stylized | Top-1 | 10.61 | 11.23 | 17.60 | 30.26 | 45.45 | 49.17 | 76.13 |
| to ImageNet | Top-5 | 22.65 | 23.08 | 34.34 | 52.52 | 69.57 | 72.30 | 92.86 |
| 10-class ImageNet | Top-1 | 15.40 | 19.00 | 25.00 | 46.20 | 75.60 | **88.40** | 77.98 |
| to Birdsong | Top-5 | 63.80 | 68.40 | 76.40 | 88.60 | 98.60 | **99.80** | 96.07 |
| Clustered-Noise | Top-1 | 4.77 | 4.11 | **96.59** | **100** | **100** | **100** | 76.13 |
| to ImageNet | Top-5 | 13.51 | 11.99 | **99.77** | **100** | **100** | **100** | 92.86 |

we introduce in Section 5.1), the weak baseline would be the performance of the ImageNet model on the Stylized data (16.4% top-5 accuracy). We choose the more competitive baseline, which is the performance of ImageNet on ImageNet data (92.86% top-5 accuracy). There is only one exception to this choice of baseline, which we emphasise when discussing the experiments. For completeness, we include the weaker baseline in Table F.1 but show the more challenging setting in the main body of the paper.

Note that since for most of the experiments in this section we are using a pretrained ImageNet model, we only do one run of the experiments. All discriminative models considered in this section are ResNet-50. All stitches in this section are trained for 20 epochs using a learning rate of $10^{-4}$ and we do not perform any hyperparameter tuning for training the stitches (full training details in Appendix B.4). All results from Section 5.1 up to and including Section 5.4 are collated together in Table 1.

### 5.1. Shape and Texture Bias

Geirhos et al. (2019) showed that standard-trained ImageNet models are biased towards learning texture information. They then constructed Stylized ImageNet (Stylized), a version of the ImageNet dataset where local texture is no longer predictive, forcing models to focus on shape information (less localised patterns).

As a real-world version of our previous shape and colour experiment, we take pretrained ImageNet and Stylized models and aim to stitch them together. Note that we cannot successfully stitch Stylized into ImageNet. As we will discuss in more detail in Section 7, this is likely due to Stylized model's poor performance on the Stylized task. Stylized only achieves 78.96% top-5 accuracy on the Stylized dataset,

which is close to the accuracy we achieve when stitching. Note, however, that we achieve stitching compatibility according to the alternative baseline (receiver's performance on the sender task), which in this case is only 16.4% top-5 accuracy. Nonetheless, we choose to consider the more challenging baseline and report stitch incompatibility instead. On the other hand, we find that **we can achieve stitch-compatibility at several different layers when stitching ImageNet into Stylized** (see Table 1). We see this as additional evidence that there are cases in which functional alignment methods such as stitching cannot distinguish between models with different biases.

### 5.2. Clustered-Noise

Following the same procedure as in the case of stitching Clustered-Noise to MNIST-trained models, we stitch Clustered-Noise to a model pretrained on ImageNet. Even with no hyperparameter tuning, we achieve stitch compatibility after the second residual block (see Table 1).

### 5.3. Different Extracted Information

We next use one dataset and stitch such that we remap its classes. This approach uses just one model and one dataset, changing fewer variables than some other experiments. We train a model on greyscale-MNIST, stitch it back together at a chosen layer, and train the stitch on MNIST data but with labels offset in a circular shift. The effect of this is that the sender's representation of, say, digit '0' is stitched so that the receiver classifies it as class '1'. Yet, an unstitched model trained on MNIST and presented with an image for '0' **must represent different information** from the same model presented with a '1' otherwise it could not discriminate. Equivalently, in the Colour-Only dataset, red would now be classified as green (see Figure A.1). This

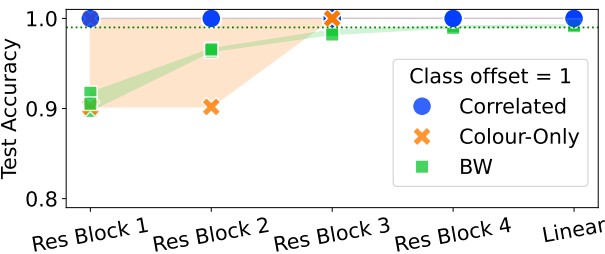

*Figure 3.* Test accuracy when stitching a model to itself with the dataset the model was trained on, but remapping class labels with an offset of 1 (e.g. an image of '7' is labelled as class '8'). Baseline performance of Correlated and Colour-Only was 100% while BW (greyscale-MNIST) was 99%, shown as green dotted line. Shaded regions show full range of results. We again find stitch compatibility, especially in the later layers.

experiment was successfully repeated for models trained on Colour-Only, and Correlated datasets (see Figure 3). The same models can be restitched with different shift offsets and at different layers (see Appendix E for more details).

### 5.4. Different Tasks and Modalities

We then attempt to stitch a model trained to recognise bird songs to a pretrained ImageNet model. For this we use a 10-class version of the xeno-canto (Vellinga & Planqué, 2015) dataset. We use publicly available code to create a version of the dataset with 10 bird classes, preprocess the data into spectrograms and train a ResNet-50 to solve the task (full training details in Appendix B.5). The model achieves 77.98% accuracy on the birdsong recognition task.

To keep in line with the typical stitching regime, we restrict the ImageNet dataset to a 10-class version. For simplicity, we choose the first 10 classes of ImageNet. In this case, we once again find we can successfully stitch models, yet their representations are not semantically similar (i.e. do not capture the same input patterns).

## 6. Stitching Other Embeddings: Autoencoders

Following Fumero et al. (2024) and Moschella et al. (2022), we consider the case of mapping representations of autoencoders. Moschella et al. (2022) extend the definition of model stitching to the generative case by stitching two autoencoders trained on MNIST from different initialisations. Their intuition is that the two models should be compatible since they are simply trained from different initialisations.

We train one autoencoder on MNIST and another one on Fashion-MNIST (Xiao et al., 2017). We take the encoder of Fashion-MNIST and the decoder of MNIST and stitch them together. We consider two approaches to stitching between inputs and reconstructions of two different datasets:

**Class Mapping** We match classes across datasets. For example, with Fashion-MNIST and MNIST, we might pair classes like 'Bag'→'0', 'Coat'→'1', etc. Within each class pair, we match images randomly. The loss is the L2 distance between the generated image and the corresponding paired image from the other dataset.

**Embedding Mapping** We match embeddings in the latent space of the autoencoders. We match the stitched embeddings of the sender against those of the original encoder of the receiver and solve this as a linear sum assignment problem. The loss is then given by the sum of squared distances between mapped embeddings. It can effectively be considered that we create a joint dataset where we pair up each training sample from dataset A (used to train encoder A) with another training sample from dataset B (used to train encoder B). Training samples are paired up using the encodings (i.e. in the embedding space). Specifically, we solve the linear sum assignment problem in the autoencoders' bottleneck to pair up samples from datasets A and B. Passing an image from dataset A through encoder A, the stitch is trained to map it to its corresponding sample from dataset B, passed through encoder B.

For full experimental details, see B.3. Following Moschella et al. (2022), we perform a qualitative evaluation of the generative stitch. We are able to reconstruct MNIST-like digits from Fashion-MNIST test data using both mappings we consider (see left-hand side of Figure 4), with clearer images obtained for the Embedding Mapping. We then stitch a CIFAR-10 (Krizhevsky, 2009) encoder onto an MNIST decoder (see right-hand side of Figure 4) and once again obtain depictions of MNIST-like digits.

Although this setting differs significantly from Bansal et al. (2021)'s original stitching proposal, it provides evidence that more broadly representations **can** be aligned without necessarily representing the same information.

## 7. Discussion and Observations

We believe building tools to meaningfully compare neural networks is an important objective in deep learning research. Such tools are treated as potential ways of better understanding neural networks and therefore can be seen as part of the Explainable AI puzzle. For example, alignment between models has been used in the attempt to better understand architectural differences (e.g. Nguyen et al., 2020; Raghu et al., 2021), aspects of continual learning (Ramasesh et al., 2021; Kim & Han, 2023), or training with different types of data (e.g. McGuire et al., 2023), or even factors that contribute to networks' alignment with human perception (Demircan et al., 2024).

Input

Embedding reconstr.

Class reconstr.

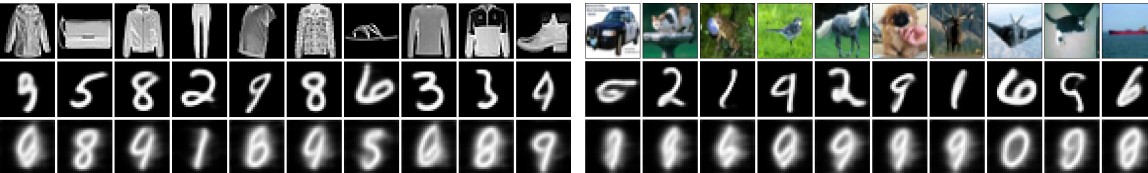

*Figure 4.* Examples of reconstructions when stitching autoencoders trained on different datasets. Left: Fashion-MNIST. Right: CIFAR-10. Top to bottom: Inputs given to the stitched autoencoders, reconstructions obtained with the Embedding Mapping method, reconstructions obtained with the Class Mapping method. MNIST-like digits are generated from Fashion-MNIST and CIFAR-10 inputs.

The objective of our paper is to more closely analyse one such tool proposed to measure similarity and alignment. From this perspective, our work fits within the literature exploring the limitations of representation comparison tools (e.g. Ding et al., 2021; Davari et al., 2023). Most related to our work but in the context of Representational Similarity Analysis (RSA), Dujmović et al. (2022) construct artificial datasets to argue that RSA cannot distinguish between models with "different feature encodings".

Through a range of experiments we have shown that we can stitch representations used to solve different tasks using different modalities. Alignments between different tasks (e.g. Gygli et al., 2021) and even different modalities (e.g. Radford et al., 2021; Merullo et al., 2023; Koh et al., 2023) are not new and have been exploited before, especially between text and images in the context of image captioning.

To explain this type of alignment, which is typically seen as an alignment of "semantic concepts", Huh et al. (2024) propose the Platonic Representation Hypothesis. This hypothesis states that models "are converging to a shared statistical model of reality in their representation spaces" which makes alignments such as stitching possible. While in the context of image captioning this argument is intuitive, Huh et al. (2024) take this hypothesis further and argue that representation alignment more generally, including across different tasks and modalities, is explained by representational convergence. This echoes the Anna Karenina Hypothesis (Bansal et al., 2021) stating that "all successful models end up learning roughly the same internal representations". Importantly, note that Huh et al. (2024) simply use functional alignment as supporting evidence for their hypothesis, but primarily focus on structural alignment.

**Are our results possible because models reached a shared understanding of reality?** No, we strongly disagree that this is applicable here. We have shown that alignments can be found between ImageNet models and birdsong recognition models, or even Clustered-Noise. It is difficult to imagine how there could be a shared understanding of reality between these three datasets.

We believe a more plausible hypothesis is that we were able to align the representations largely because they were well clustered (but not necessarily using the same information to assign samples to a cluster). While this is a straightforward observation, we include an experiment to intuitively illustrate this. We take our Clustered-Noise experiment in Section 4.2 and increase the level of noise around each cluster. As we do so, the separability decreases and, as a result, the stitching accuracy drops (see Table D.3).

Therefore, we believe our experiments cast a shadow on the interpretation of functional alignment. While it might be possible that models converge to a shared understanding of reality, we believe our results show that one needs to look beyond functional alignment to support this claim. We urge the community to rethink the belief that mapping between two clustered representations means that models capture the same aspects about the world. Models that converge to a shared statistical model of reality might be functionally compatible, but functionally compatible models do not necessarily capture the same patterns in the input.

Lastly, we believe controlled experiments where we can create models that learn to solve a task using different input cues are important for evaluating model comparison tools more generally. We regard these as necessary for establishing the validity of hypotheses around both functional and structural similarity.

**Can anything be stitched onto anything?** No. We have seen throughout the paper that some layers and even some models (Section 5.1) cannot be successfully stitched together. There are several potential reasons for this. In some cases, this could be a matter of learnability of the stitch. Note that all the experiments in Section 5 were run with the same learning rate, we did not do any hyperparameter tuning and we only trained the stitch for 20 epochs. In the case of stitching Clustered-Noise into the ImageNet model for example, it might be the case that with slightly longer training, we would be able to successfully stitch earlier layers. This highlights another limitation of stitching which is that when models cannot be stitched together, it is unclear whether this is because a good linear mapping was not found or because the representations are not linearly mappable. This is a known issue with model stitching (e.g. Hernandez et al., 2022).

The second reason why we were not able to stitch is because of the difference in task difficulty and/or original model performance and, as a result, achievable separability in the representational space. For example, in the case of stitching Stylized to ImageNet, the Stylized model achieves significantly lower performance on the Stylized dataset than the ImageNet model on the ImageNet dataset. So for the problems that the models were trained on, there is a big gap in performance. This could be because Stylized might be a harder problem to solve altogether but also because as Geirhos et al. (2019) mention, they did not necessarily train their Stylized model to achieve competitive performance on the Stylized data. It might be the case that with a better-performing model on Stylized we would be able to stitch onto ImageNet but this is beyond the scope of the paper.

**Should stitching then be used as a measure of model quality?** While there might be cases in which stitching is insightful, we believe it is difficult to think of what stitching can tell us about model quality that existing methods don't do already. If we cannot successfully stitch because of an incompatibility in task performance, this can already be seen by simply looking at the models' accuracy on the task. If two models cannot be stitched because the receiver does not cluster the representations sufficiently well at intermediate layers, then we can simply look for direct measures of representational clustering.

Finally, one of the reasons why stitching was proposed and adopted as an alternative to methods like CKA is because it allows us to identify when a representation is "better" than another, rather than simply "different". We challenge the understanding that achieving higher stitching accuracy than the baseline (receiver's own accuracy) means the representation of the sender network is better than that of the receiver. To this end, we simply swap the receivers and the senders considered in the previous experiments. In Figure 2b we observed a higher test accuracy for Colour and Correlated compared to the Digit baseline, which would be taken to indicate that their representations are "better" for discriminating samples on the problem we consider. However, stitching Digit (as sender) into Colour or Correlated (as receivers) also leads to higher accuracy for particular layers compared to the baseline (see Table C.1 and Appendix C.1), which is a contradiction. Finally, we stitch a model with itself (Digit vs Baseline) and obtain an increase in accuracy (see Figure C.1a) as well as a change in the rank of representations (see Figure C.1b). This further indicates that although the stitching layer has reduced expressivity compared to a fully-connected stitch, it still cannot be claimed that it simply aligns the representations of the sender and receiver without any additional processing. Therefore, given the contradictory results, we do not think stitching accuracy is suitable for comparing the quality of models' representations either.

**Where next?** We believe that to meaningfully compare representations we need to be able to both gauge what information they are encoding and how that information is "mapped out" in the embedding space. We have demonstrated that the functional perspective alone, and stitching in particular, cannot do that. Capturing these is of course not straightforward and the lack of ground truth makes this even more challenging. We believe the field would benefit from carefully constructing controlled benchmark problems rather than advancing based on what is "intuitively expected". Our work proposes a simple scenario for evaluating and reasoning about representational differences and similarities but it is by no means comprehensive enough for constructing a sufficiently varied baseline more generally. Nonetheless, it provides a starting point for such a framework which we believe is of paramount importance if we aim to create tools that capture meaningful differences between models.

## 8. Conclusions

We showed that networks can use or represent very different information, yet classify samples with similar accuracy. Importantly, the different representations can easily be stitched together. This leads us to question the usefulness of studying models' purely functional similarity, and in particular their stitching compatibility, to determine whether or not they capture similar information. Being able to distinguish between models that make decisions based on different input patterns is important, especially given the propensity of deep learning models to learn shortcuts. We hope that our work encourages the community to more carefully interpret the results of model stitching, and understand what it is actually responding to. We also hope our work will encourage researchers to focus on creating model comparison tools that can reliably capture informational similarity. We propose artificial shortcut learning problems as a starting point for reasoning about this in a controlled way.

## Acknowledgements

The authors acknowledge the use of the IRIDIS X High Performance Computing Facility, the ECS Alpha Cluster, and the Southampton-Wolfson AI Research Machine (SWARM) GPU cluster generously funded by the Wolfson Foundation, together with the associated support services at the University of Southampton in the completion of this work. H.M. is funded by a PhD studentship provided by the School of Electronics and Computer Science. The authors would like to thank the members of the VLC research group for useful discussions, feedback, support, snacks, and good fun.

## Impact Statement

This paper presents work whose goal is to advance the field of Machine Learning. There are many potential societal consequences of our work, none which we feel must be specifically highlighted here.

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

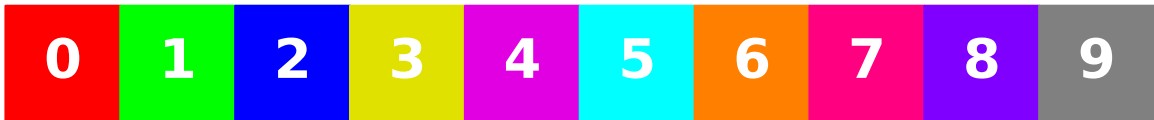

*Figure A.1.* Colour swatches for the base colour_map. Digits are font-based labels here, not MNIST images

## A. Datasets

The following variants were created:

**MNIST**   This is simply MNIST data (monochrome handwritten digits) expanded to 3 channels and normalised.

There is no target-dependent colour information for the model to learn, so we *expect* it to recognise the digits based on features such as shape, or the amount of white, or the texture of the white-grey-black edges. It *cannot* learn to use colours to recognise classes as these are never present.

It is possible that kernel weights will learn to rely on the three colour channels always being equal.

**Colour MNIST (Correlated)**  We wanted to provide an opportunity for some semantically different representations to be learned. In this case, having the background colour correlate with the target digit being displayed.

We *expect* that the models trained on Correlated data will mostly learn to rely on the colour. However, they *may* learn the shapes of the digits, or the amount of white caused by the digit. They may rely on pixels always being at one of two rgb values (white/colour) in any one image, never at an intermediate value. They may learn to rely entirely on a small number of pixels that are never white - e.g. corner pixels.

To generate the Correlated dataset, we used the *colour_MNIST* package (Bahng et al., 2020) which uses MNIST data, but changes the colour of the background depending on the digit being represented (Figure A.1). The package snaps any non-zero pixels to white ([255,255,255]). This could be thought of as changing the greyscale information to binary, and indeed reducing the information content as a result. The background colour is changed from black ([0,0,0]) to one of 10 values, always matching the data label.

For all of these datasets, to avoid the model learning something as simple as specific RGB values, the base RGB background values are modified by a random 10% per image (such that the colour is always flat, but will vary slightly from image to image within a class). The same variation is injected into the test datasets. This is analogous to requiring that colour is learned in a generalised way.

**Digit with Uncorrelated Colour (Digit)**  To encourage models to learn representations which recognise digits, but cope with different background colours, we created a dataset with digits over laid on backgrounds whose base colours were randomly selected. i.e. the digit does *not* correlate with the colour.

We expect models trained on Digit to learn features like shape. Assuming the randomisation is sufficient, they *cannot* learn to rely on colour. The choice of base colours and inclusion of colour variation mean that the Digit models *cannot* learn to rely on a single colour channel or pair of colour channels, and *must* learn to tolerate a range of colours being used. Nonetheless, they *may* learn to rely on any or all pixels in an image being one of two colours.

**Colour with Uncorrelated Digits (Colour)**  To encourage representations of colour information, but which are able to tolerate the presence of digits, we generate a dataset in the same way as for Digit but in which the background colour (not the uncorrelated digit) is the target.

Models trained on Colour *cannot* use the digits to classify because they are uncorrelated with the background colours. Successful Colour models *may* rely on colour being sampled in specific locations which never contain digit pixels (e.g. a corner), or might learn to average across the image. They may learn to rely on the presence of some white pixels as the dataset does not contain any solid-colour images. But they may not be generally immune to non-digit-like patterns of white pixels.

**Background Colour Only (Colour-Only)**  To force the learning of different representations which *cannot* be related to shape, size, or edge-effects of digits (and which may not be able to tolerate those features), the 'Colour-Only' dataset

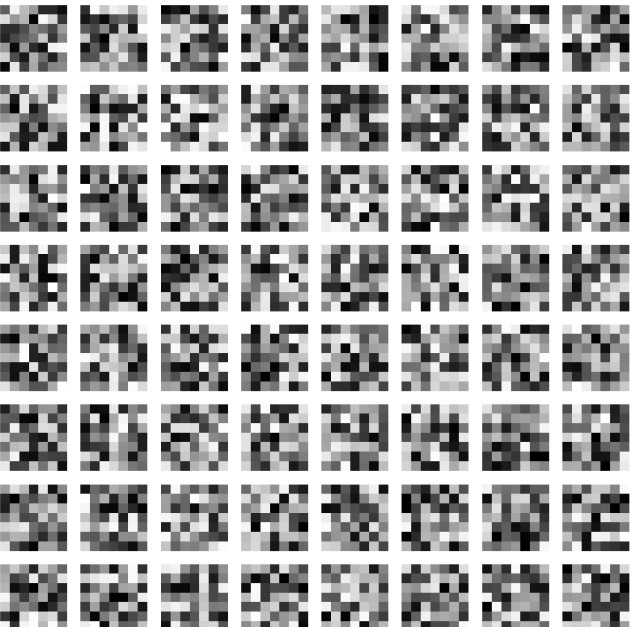

*Figure A.2.* Simulated "Clustered-Noise" data sample for ResNet-18 at Res Block 2, showing 64 7x7 feature maps. If this represented Class 0, all other Class 0 data instances would be noisy versions of this.

was created in which no digit image is present, only a solid background colour (plus 10% variation). As in 'Colour' (see Figure 2a), the base colour is the target.

The model *may* learn to rely on the colour in only one region of the image, or to assume the colour is constant everywhere.

**Clustered-Noise (Noise)** To produce a high-quality representation, without relying on specific, image-related features, we generate synthetic datasets in the form of Clustered-Noise.

Passing an image through the first layers of a sender model produces a set of activations specific to the final layer before the cut. For example, with ResNet-18 cutting just before Res Block 2 of the receiver, the sender output will be 64 channels of size 7x7 (Figure A.2.) The synthetic dataset must be matched in shape to the layer it is being stitched at. As such, it has to be regenerated depending on the layer in question. It does not produce images, rather synthetic activations. Each data instance is represented by a point in activation space based on its class, plus a random offset vector. Thus each class is located in a cloud or $\epsilon$-ball. For example, in the Res Block 2 example, each of the ten classes will be centred around a different random point in 3136-dimensions.

There is no attempt to create structure within or between the feature maps, and it is not derived from actual activation data. The high-dimensionality means it is likely that the clusters of data points will be highly separable, though this is not enforced and has not been verified.

To ensure the same base representations are used for train and test datasets for a given layer or test, a `generate_activations()` function was created (Alg 1). A train and test dataset can then be generated (Alg 2) and accessed via a dataloader.

## B. Experimental details

### B.1. Stitching Between Models

In an extension of the experiments by Bansal et al. (2021), we train models on different special datasets (Section A) and then stitch between them, assessing the change in accuracy.

**Hyperparameters for Model Training** batch_size=128, 4 Epochs, SGD, lr=1e-1, momentum=0.9, weight_decay=1e-4.

---

**Algorithm 1** Generate Base Activations

---

**Procedure** GENERATE_ACTIVATIONS($num\_classes$, $representation\_shape$)
**for** $c \leftarrow 0$ **to** $num\_classes - 1$ **do**
   $activations[c] \leftarrow rand(representation\_shape)$
**end for**
**return** $activations$
**End Procedure**

---

**Algorithm 2** Generate a dataset (unshuffled)

---

**Procedure** SYNTHETICDATASET($train$, $activations$, $noise$)
**if** $train$ **then**
   $SamplesPerClass \leftarrow 6000$
**else**
   $SamplesPerClass \leftarrow 1000$
**end if**
**for** $c \leftarrow 0$ **to** $num\_classes - 1$ **do**
   $data[c * SamplesPerClass : ((c + 1) * SamplesPerClass)] \leftarrow activations[c]$
   $targets[c * SamplesPerClass : ((c + 1) * SamplesPerClass)] \leftarrow c$
**end for**
$data \leftarrow data + noise * randn$
$data \leftarrow clamp(data, 0, 1)$
**End Procedure**

---

**Hyperparameters for Model Stitching** batch_size=128, 10 Epochs, SGD, lr=1e-4, momentum=0.9, weight_decay=1e-2

**Hyperparameters for Noise Stitching** batch_size=64, 4 Epochs, SGD, lr=1e-4, momentum=0.9, weight_decay=1e-2

Two versions of this are performed, one stitching from each different model into the Digit with Uncorrelated Colour receiver model, and the other stitching from that model as sender into each of the others. This will allow us to assess the symmetry of the stitching process. Recall that models trained on each of the datasets represents a model that *could* have arisen naturally by training on Correlated data, but with the advantage that we know something about features that may (or cannot) be learned.

In this experiment, all image datasets are used to train models. Each of the models (including Digit) is then stitched into the Digit model at each of 5 points: (e.g. see Figure B.1).

- Cut before Res Block 1

- Cut before Res Block 2

- Cut before Res Block 3

- Cut before Res Block 4

- Cut before Linear Layer

The accuracy and rank are then measured using the Correlated test dataset. Accuracy and rank are also measured for the whole, unstitched 'Digit' model using the Correlated test dataset for comparison. This test configuration is different from that used by Bansal et al. (2021) and Hernandez et al. (2022). They used the same dataset for training and testing their models. We wanted to examine the case where:

1. The dataset for training and testing the stitch is correlated (biased) - i.e. it contains class-correlated features other than the intended learning target (the written digit): in general this will be the case, even if unintentionally. Recall that the bias is that the background colour is correlated with the labelled digit.

2. The sender models are *likely* to have learned different features from the receiver.

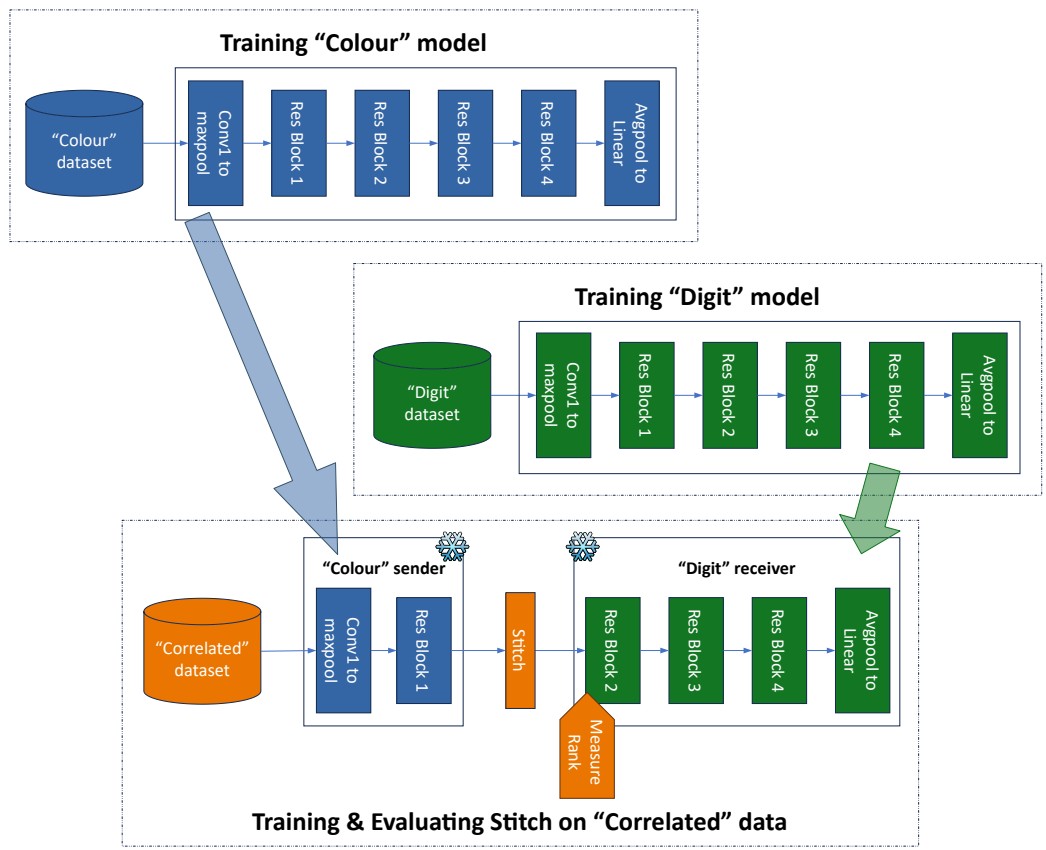

*Figure B.1.* To illustrate the process, prior to any stitching, one model is trained on Colour dataset and another on the Digit dataset. Because it will be used a receiver, the Digit model is then evaluated on the Correlated dataset as a baseline, without any stitch layer. For stitching, the Correlated dataset is used: the train dataset for training the stitch, and the test dataset for evaluating functional performance. In this example, the Colour model is the sender and the cut is before Res Block 2. The Digit model is the receiver and the stitch layer connects just after its Res Block 2. Note that the Rank is measured at the first conv layer of the Digit receiver model

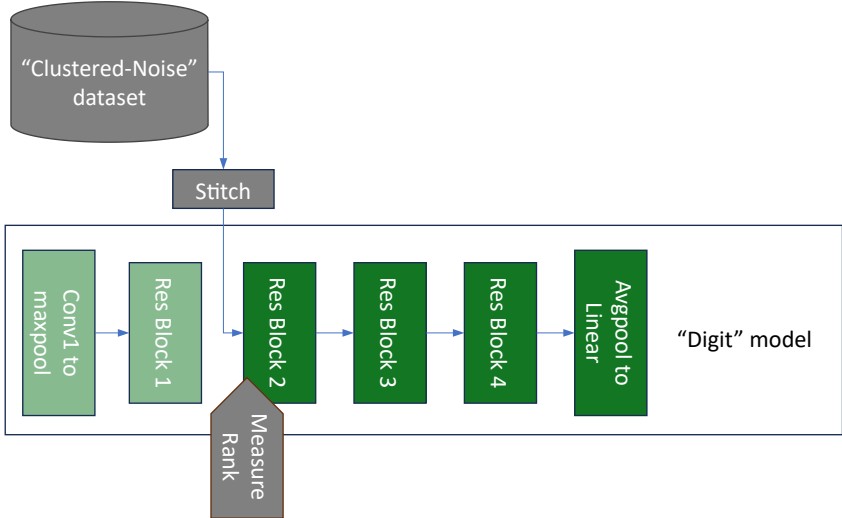

*Figure B.2.* In this example, the Digit model is the receiver and the cut is before Res Block 2. Note that the Rank is measured at the first conv layer of the Digit receiver model. The shape of the 'Clustered-Noise' data must match the receiving layer.

## B.2. Stitching from Clustered-Noise

We train ResNet-18 models for each of the image-based datasets described. For each model, the accuracy is recorded for the associated test dataset. The model is cut before the first ResNet block, and prepended with the stitch to create a stitch + receiver model. The stitch (only) is then trained on the Clustered-Noise dataset.

The stitch + receiver model (Figure B.2) is then tested against the synthetic test dataset. This is repeated, cutting before each Res Block (a layer comprises 2 Basic Blocks. The skip connections are preserved), and before the final fully-connected linear layer. We also record the rank of the activations for the Clustered-Noise test dataset at the first convolutional layer in the receiver model after the stitch. This is to provide insight into how the stitched data is "perceived" by the different models. Also how that compares to when dataset images are processed by the model. For each test at each stitch point the synthetic dataset was regenerated to reduce the likelihood that results are due to one randomly selected set of points having a significant structure.

### B.2.1. EXAMINING THE RANK

To gain further insight into how the representations being sent are "perceived" by the receiver, we analyse the rank at the first receiver layer. Note, however, that generally analysing the rank of two sets of representations cannot tell us anything meaningful about their informational similarity. We are simply arguing that the receiver cannot map the various sender representations in an entirely equivalent way if the rank in the first receiving layer is different.

1. Train several models each on a different, but related dataset. For example, Colour MNIST can create different dataloaders of MNIST digits on coloured backgrounds where the colours correlate or do not correlate with the digit class.

2. Stitch models to each other such that sender is from layer 1 to L, and receiver is from layer L+1 to Classifier. Train the stitch.

3. Collect the representations at layer L+1 obtained using the test dataset. For estimating the rank, we follow Masarczyk et al. (2023), who compute the singular values of the sample covariance matrix and threshold these at 1e-3 of the largest singular value.

## B.3. Stitching Autoencoders

We tested whether two autoencoders, trained on different datasets, could be stitched at their bottlenecks. In our experiments the stitching is placed immediately after the encoder. Autoencoders are typically trained using reconstruction loss, usually

based on L1 or L2 distance between input and output images. However, when the input and output images come from different datasets, reconstruction loss becomes inappropriate since the images won't match. To address this, we use two methods for learning to map images from one dataset to another, we refer to these approaches as class mapping and embedding mapping.

**Class Mapping** When datasets have class labels, we can match corresponding classes across datasets. For example, with CIFAR-10 and MNIST, we might pair classes like 'Airplane'→'0', 'Horse'→'1', etc. Within each class pair, we match images randomly. During training, an image from one dataset is passed through its encoder, then through the stitch, and finally through the decoder corresponding to the other dataset to generate an image. Our loss function is then the L2 distance between the generated image and the image that has been paired with our input image.

**Embedding Mapping** Alternatively we can match points one-to-one in the latent space of the autoencoders and train the stitch to map points from one latent space onto points of the other. This involves first encoding batches from both datasets through the respective encoders, producing features $e_1$ and $e_2$. The encodings $e_1$ are then passed though the stitch, producing features $s$. We then match points $s$ to points $e_2$ in a one-to-one manner such that the sum of the distances is minimised. This mapping challenge is known as the linear sum assignment problem, which can be efficiently solved using SciPy. Once this mapping has been found, our loss function is simply the sum of squared distances between paired points in $s$ and $e_2$. The full calculation of this loss is detailed in Algorithm 3.

---

**Algorithm 3** One-to-One Autoencoder Stitching (Loss Calculation)

---

**Procedure** CALCULATE_LOSS($AE_1$, $AE_2$, $stitch$, $dataset_1$, $dataset_2$)
$d_1 \sim dataset_1$
$d_2 \sim dataset_2$
$e1 \leftarrow AE_1.encoder(d_1)$
$e2 \leftarrow AE_2.encoder(d_2)$
$s \leftarrow stitch(e_1)$
$cost\_matrix \leftarrow PairwiseDistances(s, e_2)$
$i, j \leftarrow LinearSumAssignment(cost\_matrix)$
**return** $\sum cost\_matrix[i, j]$
**End Procedure**

---

We trained autoencoders to map Fashion-MNIST to MNIST and CIFAR-10 to MNIST using these techniques. The architecture of our encoder includes convolutional layers, ReLU activations, and max pooling layers for downsampling. Following this, features are flattened and passed through a series of linear layers. The decoder follows the same structure but in reverse. The bottleneck is a vector of length 128 for all experiments. All parameters were frozen except for the stitch, which connected the encoder of one model to the decoder of the other. All autoencoders were trained for 25 epochs with a learning rate of 1e-4 using the Adam optimiser. Embedding Map stitching was trained for 25 epochs using SGD with a learning rate of 1e-5, momentum of 0.9, and weight decay of 0.01. The class mapping stitching was trained for 20 epochs using SGD with the same parameters except the learning rate was set to 1e-2. The learning rate and total epochs for each setting were tuned manually through analysis of the corresponding loss curves.

Results can be seen in Figure 4. We can see that MNIST-like digits can be reconstructed even when inputs are from a different dataset. Class Mapping stitching tends to produce blurrier images. This is likely because, within each class pairing, images are matched randomly, which leads to an output that is essentially an average of all images in the class. Embedding Mapping stitching can produce cleaner MNIST images.

### B.4. Stitching on Real-world Datasets

All stitches in Section 5 are trained for 20 epochs using SGD with learning rate and weight decay of $10^{-4}$, momentum of 0.9, batch size of 128.

### B.5. Training for Birdsong Recognition

We use publicly available code to train a ResNet-50 for birdsong recognition. We start with a random initialised ResNet-50 (unlike the original code) and use SGD to train for 20 epochs. We perform a hyperparameter search over the learning rate

*Table B.1.* Results of stitching two ResNet18 model together at different layers with different amounts of regularization $\lambda$. Model $A$ was trained to predict the background colour on Colour-MNIST, whereas model $B$ was trained to predict the digit.

| Sender Model | Receiver Model | $\lambda$ | Block 1 | Block 2 | Block 3 | Block 4 | Block 5 | Linear |
|:---:|:---:|:---:|:---:|:---:|:---:|:---:|:---:|:---:|
| | | 0.01 | 99.90% | 99.96% | 100.00% | 100.00% | 100.00% | 100.00% |
| $A$ | $B$ | 0.1 | 96.06% | 99.09% | 99.83% | 59.96% | 66.07% | 99.92% |
| | | 1 | 53.43% | 22.23% | 49.04% | 96.66% | 9.74% | 9.74% |
| | | 0.01 | 100.00% | 100.00% | 100.00% | 100.00% | 100.00% | 100.00% |
| $A$ | $A$ | 0.1 | 98.74% | 93.12% | 88.16% | 85.74% | 23.07% | 88.97% |
| | | 1 | 54.12% | 38.46% | 81.59% | 86.72% | 10.28% | 10.28% |

($10^{-4}$ to $10^{-1}$), weight decay ($10^{-4}$ to $10^{-3}$) and momentum values (0.8 to 0.9). We believe that tuning the hyperparameters further would neither increase nor decrease the support for the point we are trying to make. If model stitching is a reliable model comparison method, it should work for a range of models from high-performing ones all the way to models trained without hyperparameter tuning.

For training the final model the following configuration was used: learning rate of $10^{-2}$, weight decay of $10^{-4}$ and momentum of 0.9. Note that in the original manuscript we accidentally reported a higher baseline for the birdsong model. However, in terms of layer stitch compatibility, all results remain the same (top-1 compatibility starts before the linear layer and top-5 compatibility starts before the 5th residual block).

### B.6. Regularizing the Stitch

To explore different types of stitching, we experiment with training a stitching with additional regularization. The first network $A$ is a ResNet18 trained on Colour-MNIST digits, but with the label corresponding to the background colour. The second network $B$, also a ResNet18, is trained on the same images, but now the labels correspond to the digit instead of the background colour. We will stitch these networks together with a stitch $S$ at layer $l$ such that $A$ is our sender network and $B$ is our receiver. For the sake of comparison we will also look at stitching network $A$ into itself.

These stitches will be trained using BCE loss with an additional L1 regularization term on the parameters of the stitch. So our loss for this will be:

$$\mathcal{L} = \mathcal{L}_{BCE} + \lambda\|\theta_S\|_1, \tag{1}$$

where $\mathcal{L}_{BCE}$ is our original loss function, $\lambda$ is a scalar controlling the strength of our regularization, and $\theta_S$ are the parameters of the stitch. Results of additional L1 regularization are shown Table B.1 where Block 3, Block 4, and so on refer to the layer at which the stitching took place. Percentages refer to accuracy of the final model. When our regularization strength is $\lambda = 0.01$ we are still able to stitch $A$ into $B$. This is not true for larger values of $\lambda$, but larger values also prevents $A$ being stitched into itself, suggesting that the regularization is too strong. In practice, we see that even for $\lambda = 0.01$, the regularization term still dominates the loss. This demonstrates that even with relatively strong L1 regularization our stitching results are still valid.

We repeat this experiment with weight decay and $\lambda$ both set to 0. Weight decay is L2 regularization, so this setting corresponds to training the stitch with no regularization at all. For this setting we achieve above 99% stitching accuracy for all layers, suggesting that our results are robust across different types of stitches, both with and without regularization.

## C. Different Senders to Digit Receiver under Various Architectures

### C.1. ResNet-18: Stitching to Digit

As mentioned in Section 7 and Appendix B we used the Digit with Uncorrelated Colour (Digit) model as both receiver and sender in the stitch, and also used Clustered-Noise as a synthetic sender model. Here we present those extended results.

In Table C.1 we can see that the base accuracy of the Digit network is 0.978. When stitching Colour-Only into this, accuracy improves to 0.999, indicating that Colour-Only is "better than" Digit. However, the base accuracy (against the same test set

*Table C.1.* Accuracy of trained ResNet-18 models against 'Correlated' dataset (average of 5 initialisations).

| Model | Base Accuracy | Acc. with stitch before Res Block 1 | |
| | | 'Digit' is Receiver | 'Digit' is Sender |
| --- | --- | --- | --- |
| Correlated | 0.999 | 0.999 | 0.981 |
| Digit | **0.978** | 0.995 | 0.996 |
| Colour | 1.000 | 0.999 | 1.000 |
| Colour-Only | **0.686** | **0.999** | **0.985** |

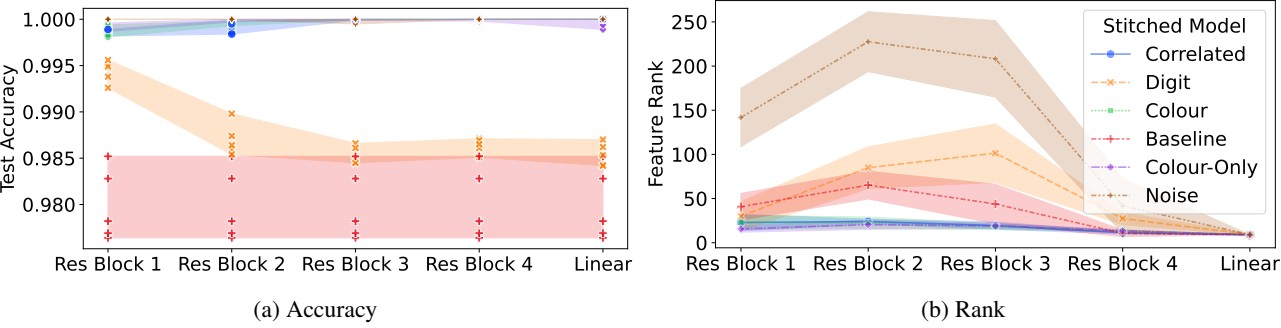

|  (a) Accuracy  |  (b) Rank  |

*Figure C.1.* **(a)** Test accuracy on Correlated data when the 'Noise', and each of the trained ResNet-18 models are stitched into the Digit receiver (baseline is the accuracy of the original Digit model on test Correlated data). This extends Figure 2b by including Colour-Only and Clustered-Noise. Shaded areas cover 100% of results to highlight the variability. **(b)** Rank analysis of the stitched models' representations. Shaded areas mark 1 standard deviation.

of Correlated) of the Colour-Only network is 0.686, but when stitching Digit into it as a sender, the accuracy improves to 0.985, indicating that Digit is "better than" Colour-Only. This contradiction constitutes a problem for the claim that model stitching can be used to ascertain relative quality. Note also that stitching the Digit model into itself produces a performance improvement from 0.978 to 0.995 which challenges the belief that the $1 \times 1$ convolution with batchnorms stitch (Bansal et al., 2021) does not add capacity.

In Figure C.2 we illustrate the fact that for each initialisation the stitched Digit model outperforms the unstitched baseline. Shown on the plot are the highest baseline to the lowest stitch accuracy in one model. Baseline = 0.985, Acc_min = 0.986. Also show is lowest accuracy of any stitch in any experiment from its baseline. Baseline = 0.978, Acc_min = 0.984. All stitched Correlated and Colour models perform above all baseline results.

## C.2. VGG19: Stitching to Digit

In a variant of the ResNet-18 experiment from 4.1 we stitched from differently biased VGG19 models into ones trained on Digit data, using Correlated data to train and test the stitch (repeated for 4 differently initialised models). This is indicative of comparing models which have learned different biases and are evaluated on biased data. As can be seen in Figure C.3 most models show approximately equivalent, or improved performance at all stitch points even though the information represented is different.

We believe that being unable, sometimes, to stitch differently biased models is irrelevant: the issue is that if it is sometimes possible to form such a stitch then one cannot use stitch-compatibility to identify equivalent information.

## C.3. LeNet-like Models: Stitching to Digit

During the rebuttal period we started extending the stitching experiments from Section 4.1 with a LeNet-like architecture compatible with $28 \times 28$ input size. Note that we only train the stitch for 10 epochs so more stitch compatibilities could be found. However, for the purpose of our paper it suffices to show that stitching compatibility exists. Results are presented in Table C.2

Note that during the rebuttal we also tried stitching randomly initialised senders in the LeNet-like case. For the setting we

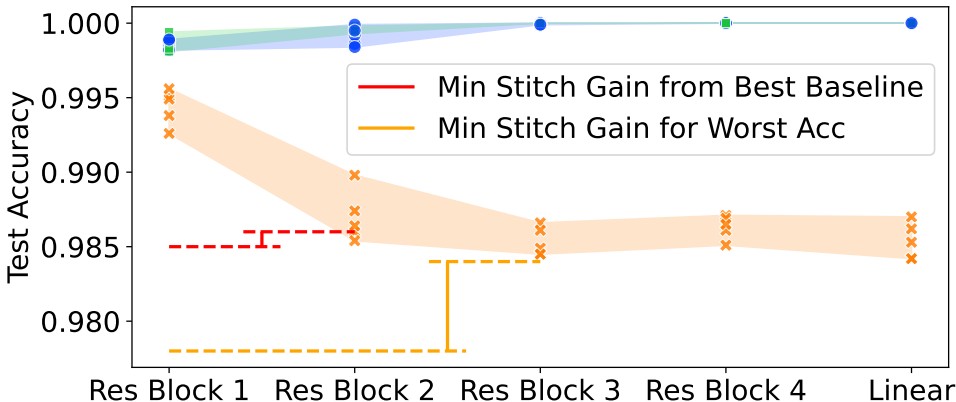

*Figure C.2.* As in Figure 2b we show the same Correlated, Colour and Digit results. We mark the stitched performance compared with baseline as Stitch Gain (i.e. negative Stitch Penalty) for two cases.

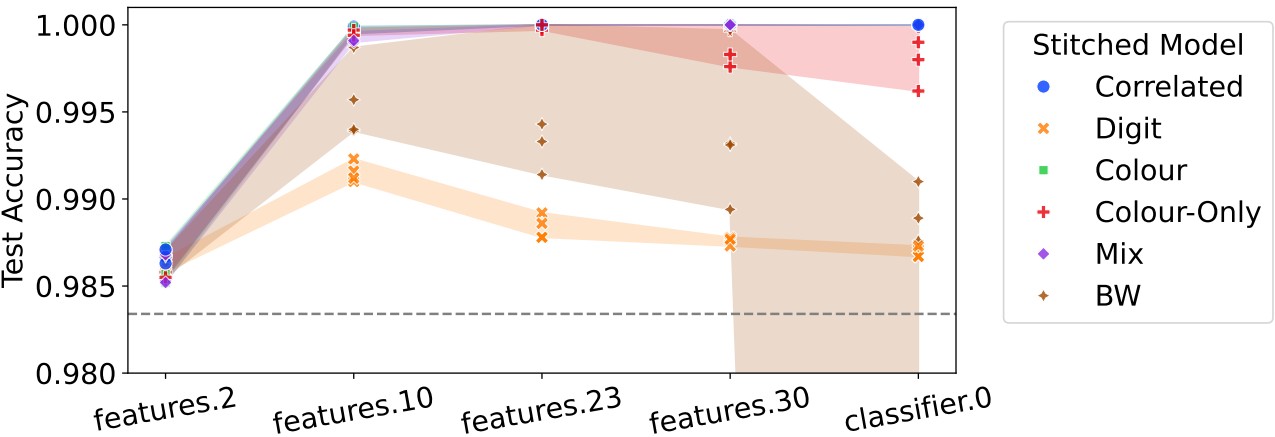

*Figure C.3.* Accuracy of variously trained VGG19 Sender models stitched into Digit model at 5 different points in the network using Correlated dataset. Dashed grey line shows baseline performance of whole Digit model against Correlated dataset. Shaded areas cover 100% of results to highlight the variability. Mix is a dataset comprising examples from MNIST, Correlated, and Colour-Only. BW is greyscale-MNIST

*Table C.2.* Accuracy (%) when stitching the model trained purely on colour patches (Colour) and the model trained on correlated digit and colour (Correlated) with a receiver trained on grayscale MNIST (Digit). Stitching compatibility can be easily found when stitching at at least one layer.

| | Conv 1 | Avg Pool 1 | Conv2 | Avg Pool 2 | Reference |
|---|---|---|---|---|---|
| Colour to Digit | 76.94 | 76.68 | **97.94** | **97.34** | 88.19 |
| Correlated to Digit | 79.11 | 78.66 | **93.24** | **92.90** | 88.19 |

*Table D.1.* Accuracy (%) of trained models against same test dataset (average of 3 initialisations). Values shown to aid reading of Figure D.1a

| Model | Test Accuracy |
|---|---|
| 'Correlated' | 100 |
| 'Digit' | 98 |
| 'Colour' | 100 |
| 'Colour-Only' | 100 |

ran, we found that the trained sender was successfully stitching, whereas the randomly initialised sender failed to stitch. Re-running the experiment with different seeds after the rebuttal period we found that there were cases where randomly reinitialising also led to a successful stitch. See Appendix G.5 for a discussion on stitching with randomly initialised senders.

## D. Further Experiments with Clustered-Noise

### D.1. VGG19 Results for Clustered-Noise Senders

There is a reasonable question about whether ResNet architectures may respond to model stitching in a particular way because of the skip connections. To address this, we extended the testing with Clustered-Noise data to include VGG19 models. 3 initialisations were run. 50 Epochs for model and stitch training: batch_size=64, SGD, lr=1e-2, momentum=0.9, weight_decay=1e-4.

Each trained model was cut and stitched at the following points. We decided to take a sample rather than testing every convolutional layer to reduce experimental duration (Figure D.1a):

- Whole Model (image data)

- Cut before features.2 (Noise data)

- Cut before features.10

- Cut before features.23

- Cut before features.30

- Cut before classifier.0

We record the rank of the activations for the Clustered-Noise test dataset at the first convolutional layer in the receiver model after the stitch (Figure D.1b).

Figure D.1a and Table D.1 show that 'Digit' has improved results with stitching in the 'Clustered-Noise' dataset. 'Correlated' maintains accuracy. This is in accordance with the results for ResNet.

Most notably, 'Colour' and 'Colour-Only' show very varied results with stitching. It is not clear from this limited experiment whether that is due to the network not being stitching-compatible with the synthetic Clustered-Noise data, or if it is a manifestation of the random variability shown by Csiszárik et al. (2021) (i.e. stitching from different stitch initialisations can yield very different stitching-penalties).

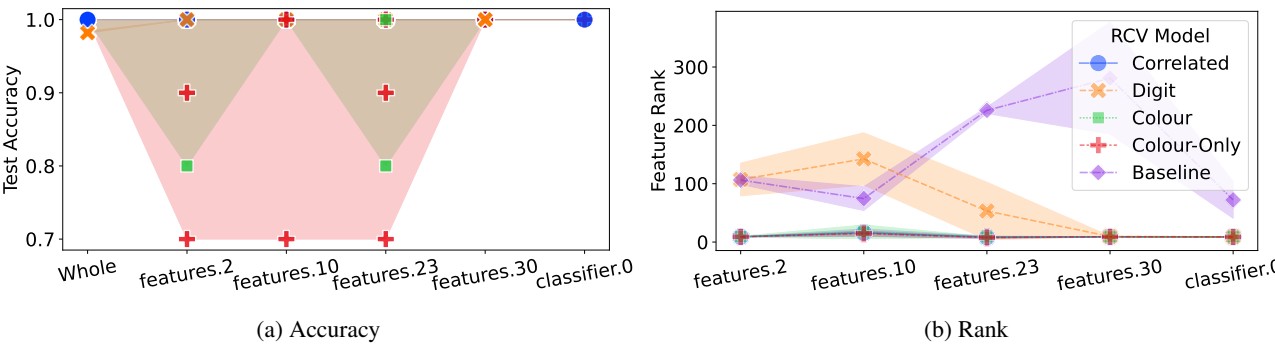

(a) Accuracy            (b) Rank

*Figure D.1.* **(a)** Accuracy of VGG19 models trained on the four dataset variants when Clustered-Noise is stitched at each of 5 different points in the model. The first sample shown ('Whole') is for the uncut models tested with their own test datasets. Shaded areas cover 100% of results to highlight the variability. Markers show individual measurements. **(b)** Ranks of activations just after the stitch from Clustered-Noise test dataset. 'Baseline' shows ranks at the same points of the uncut 'Digit' trained VGG19 models when presented with 'Digit' test data. Shaded areas mark 1 standard deviation.

We carried out a small investigation using the model which performed worst at features.2 stitching. Trying multiple stitch initialisations gave accuracies in the range $\sim 40\% - 80\%$. Trying multiple Clustered-Noise dataset initialisations gave similar results. Lower and Higher learning rates were also tried with no change. It may be that the training parameters are wrong as sometimes a good stitch-training loss increases between epochs, and we did not optimise stitch training hyperparameters. However, it may be that some models reach points in the loss landscape which it is hard to match when stitching from 'Noise' data.

This may represent a difference between ResNet-18 and VGG19 architectures. Two possible reasons (which deserve to be investigated) are:

**Skip Connections** ResNet skip connections may make stitching easier.

**Dimensionality** VGG19 at features.2 has 64 feature maps of size 32x32, whereas Res Block 2 of ResNet-18 has 64 feature maps of size 7x7. The 1x1 Convolutional stitch layer can only learn to create linear combinations of its input feature maps and it may be harder to create necessary patterns of activation when (in VGG19) the maps have $\sim 21$ times as many units.

The most obvious feature of the rank information in Figure D.1b is the difference between ranks in the unstitched 'Digit' network (presented with 'Digit' data) and the stitched networks presented with 'Noise' data. This demonstrates that being stitched to a sender can produce similar functional results in terms of overall accuracy, while internal information is different.

### D.2. ResNet-50: Stitching 'Clustered-Noise' into ImageNet

It can be argued that Colour MNIST-trained ResNet-18 models can be stitched from 'Noise' only because they are simple. To investigate this, we use ResNet-50 from torchvision pretrained on IMAGENET1K_V1. The model is cut after the avgpool layer, before the final fully-connected linear layer, and prepended with the stitch to create a stitch + receiver model. The stitch (only) is then trained on the 'Noise' dataset, generated for 1000 classes. The stitch + receiver model is then tested against the synthetic test dataset. This is repeated, cutting before other ResNet-50 Res Blocks (Table 1). At each Res Block from 3 onwards, accuracy exceeded the ImageNet baseline.

The experiment was repeated for some of the later layers for IMAGENET1K_V2 (see Table D.2). Top-1 results only. Again, at these layers stitch performance exceeded the baseline (80.86%)

### D.3. ResNet-18: Increasing the Cluster Size of 'Clustered-Noise'

In the 'Noise' stitching experiments described above (Appendix B.2, D.1, D.2), the radius of each cluster was 0.1, resulting in tightly grouped class representations which are likely to be separable. We hypothesise that it is this separability which leads to high performance when stitching. To investigate, stitching was repeated with ResNet-18 for one MNIST and one

*Table D.2.* Accuracy of ImageNet pretrained ResNet-50 models (IMAGENET1K_V2) against 'Noise' dataset. Numbers of epochs adjusted to reduce processing time.

| Cut before layer | Acc. (%) | Epochs |
|---|---|---|
| Res Block 3 | 99.8 | 3 |
| Res Block 4 | 100 | 3 |
| AvgPool | 100 | 10 |
| Linear | 100 | 10 |

*Table D.3.* Varying the radius of noise clusters at one stitch point for two trained models. Original model accuracy = MNIST: 98%, Colour-Only: 100%

| | Acc. (%) | |
|---|---|---|
| Radius | MNIST | Colour-Only |
| 0.5 | 99.7 | 99.8 |
| 0.9 | 97.3 | 97.8 |
| 0.99 | 96.4 | 96.3 |
| 1.0 | 96.0 | 96.6 |

Colour-Only model cut before Res Block 3 for varying noise radii. For each model, the cluster centre positions were kept consistent while the data points around them were regenerated. Increasing the Clustered-Noise radius decreased accuracy (see Table D.3) meaning that it is possible to tune noise to have a stitching penalty above, below, or matching the original network. We argue that representational cluster shape may account for the quality of stitches between real-world networks.

### D.4. Misaligned Class Size

During the rebuttal period we experimented with stitching a smaller number of clusters (classes) into the pretrained Stylized and ImageNet receivers. We considered 10, 5, and 2 random noise clusters and successfully managed to stitch these into the receiver. As expected, the fewer classes represented, the easier it was to achieve stitch compatibility. In the case of stitching two classes to the ImageNet model, for example, 100% accuracy was achieved within only one epoch of training.

## E. Different Offsets for Remapping Classes

In subsection 5.3 we describe experiments in which a trained network is cut and self-stitched; the original model's dataset is used to train the stitch, but with an offset in the class labels. We present and discuss further results here.

Performance was sometimes reduced in earlier layers (see Figure E.1) for some datasets, but note that Colour-Only could sometimes achieve perfect performance even when stitching before Res Block 1.

BW (greyscale-MNIST) always suffered some degradation at early layers, but later in the network was successful. As a short investigation, we ran the MNIST offset tests for a single model at all 10 offsets and found similar curves for all offsets other than zero. We hypothesise that this is due to different classes having very varied clustering at early layers, but investigating this is beyond the scope of this paper.

## F. Alternative Baselines

As described in section 5, we present stitching data with reference to both baselines in Table F.1. Note that in the main body of the paper we choose the most competitive between the two. While this is usually the performance of the receiver network on the receiver task, there exists one exception. Stylized performs better on ImageNet than on Stylized and therefore we choose the performance on sender's task as the baseline. Note that with the alternative baseline we observe full stitch compatibility.

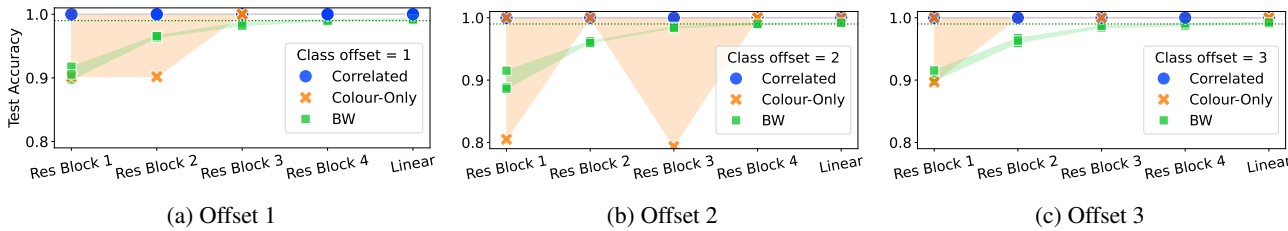

(a) Offset 1                  (b) Offset 2                  (c) Offset 3

*Figure E.1.* Test accuracy shown for three initialisations of three datasets stitching before each of 5 layers and for different class offsets. Shaded regions cover 100% of results to highlight variability

*Table F.1.* Results of stitching various models into and from ImageNet. We highlight in bold the representations that were stitch-compatible. Note that "Block $X$" and "Linear" means that we are stitching **before** the $X$th residual block and the classification layer respectively. Sender Baseline is the performance of the receiver on the given task (sender's task), while Receiver Baseline is the performance of the receiver on the receiver task. *These are the accuracy measurements we obtain when resizing the test samples to 256, as described in the paper. We can only match (Geirhos et al., 2019)'s reported top-5 accuracy of 16.4% (with a corresponding 73.8% top-1 accuracy) when removing the resize operation.

|  |  | Block 1 | Block 2 | Block 3 | Block 4 | Block 5 | Linear | Sender Baseline | Receiver Baseline |
|---|---|---|---|---|---|---|---|---|---|
| ImageNet to | Top-1 | 58.39 | 58.61 | **61.53** | **65.31** | **70.43** | **74.34** | 60.18 | 56.18 |
| Stylized | Top-5 | 80.57 | 80.61 | **83.28** | **86.13** | **89.78** | **91.93** | 82.60 | 78.96 |
| Stylized | Top-1 | 10.61 | 11.23 | 17.60 | 30.26 | 45.45 | 49.17 | 7.15* | 76.13 |
| to ImageNet | Top-5 | 22.65 | 23.08 | 34.34 | 52.52 | 69.57 | 72.30 | 15.87* | 92.86 |
| 10-class ImageNet | Top-1 | 15.40 | 19.00 | 25.00 | 46.20 | 75.60 | **88.40** | 6.59 | 77.98 |
| to Birdsong | Top-5 | 63.80 | 68.40 | 76.40 | 88.60 | **98.60** | **99.80** | 50.40 | 96.07 |
| Clustered-Noise | Top-1 | 4.77 | 4.11 | **96.59** | **100** | **100** | **100** | – | 76.13 |
| to ImageNet | Top-5 | 13.51 | 11.99 | **99.77** | **100** | **100** | **100** | – | 92.86 |

## G. Comparison with Randomly Initialised Sender Networks

As part of the evaluation of the potency of the stitching architecture, Bansal et al. (2021) examined the performance when stitching from a randomly initialised sender network, showing that stitching penalty increases (accuracy decreases) as one stitches at later points in the networks.

Importantly, note that **this is not given as a precondition** for the application of stitching to model comparisons. Bansal et al. (2021) simply perform this experiment for one of the many architectures they consider as a way of confirming that the stitching layer "is not performing learning". Nonetheless, we examined it in some of our settings. We conclude this series of experiments with a discussion (Appendix G.5) on how they align with the perspective presented in this paper.

### G.1. ResNet-18: Randomly Initialised Sender to Digit

We extend the setting of Figure C.1 by attempting to stitch randomly initialised (untrained) networks into a Digit receiver using Correlated dataset. This was repeated for the same receiver but using a different seed for the random network and stitch process (see Figure G.1).

As was seen in Figure C.1a, stitching improves performance relative to the baseline. The randomly initialised models perform better than the Digit model, but less well than Correlated, Colour, Colour-Only and Clustered-Noise.

### G.2. ResNet-18: Randomly Initialised Sender to BW Receiver with Colour-Only Dataset (and to Colour-Only Receiver with BW Dataset)

In subsection 4.2 we indicated that using Colour-Only senders and stitch training datasets with Greyscale-MNIST receivers results in 100% accuracy. For comparison, we tested for a single Greyscale-MNIST receiver stitching a randomly initialised

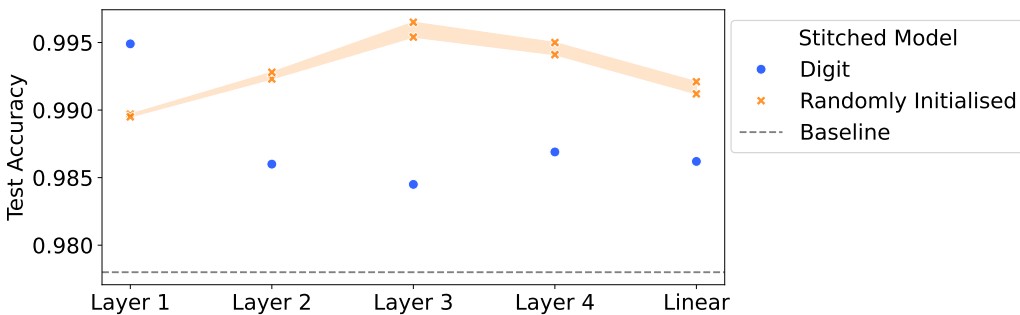

*Figure G.1.* ResNet-18 trained on Digit compared with stitching from two randomly initialised sender networks. Stitch training and evaluation used Correlated dataset. Digit plots are for self-stitching. Baseline is whole Digit model tested with Correlated dataset

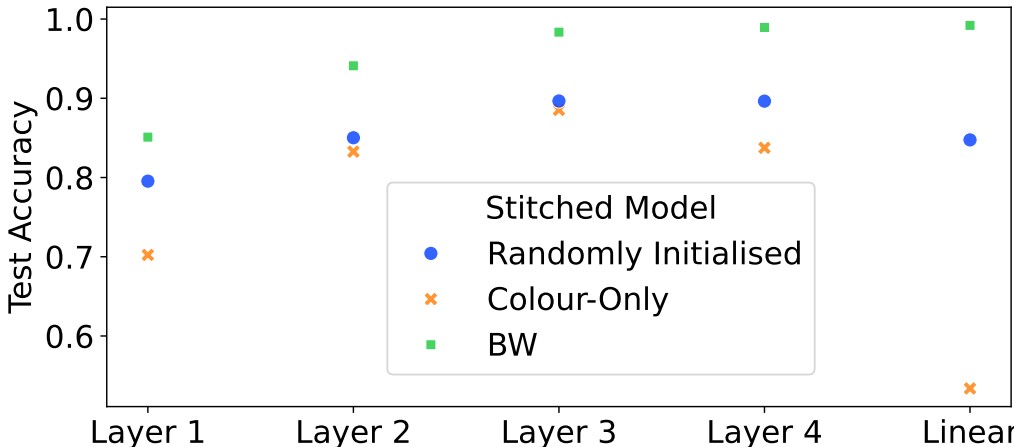

*Figure G.2.* ResNet-18 trained on Colour-Only stitching from a randomly initialised sender network - compared with a sender trained on BW data. Stitch training and evaluation used Greyscale-MNIST (BW) dataset. Also shown is self-stitching with Colour-Only as its own sender, also on BW dataset.

sender with Colour-Only data and found that it also achieved 100% accuracy at each layer. The same result was obtained self-stitching the same Greyscale-MNIST model as its own sender.

We also did the complementary experiment Colour-Only receiver, and Greyscale-MNIST datasets for the stitching (Figure G.2). The Randomly initialised model underperforms relative to the BW sender model as expected, given that the BW-trained sender model should be able to separate BW data. It also outperforms the Colour-Only sender stitched into itself as a receiver.

### G.3. VGG19: Randomly Initialised Sender to Digit with Digit Dataset

For an arbitrarily selected VGG19 model trained on Digit dataset, we found baseline performance was very similar to stitching the network to itself at each layer (Figure G.3). By comparison, stitching with Digit dataset from an uninitialised sender shows performance degrades with the depth of stitch as expected.

Varying the experiment to more closely mirror the ResNet-18 work in Figure G.1 and the VGG19 work in Figure C.3, we find that training the Digit stitch on the Correlated dataset gives an improvement in performance relative to the baseline of Digit model tested on Correlated dataset. However, this is outperformed by the uninitialised sender (see Figure G.4).

### G.4. Linear Probing Randomly Initialised Senders

To give more perspective on the results obtained in this section, we carry out an additional set of experiments where we verify the level of linear separability of representations obtained by passing datasets through randomly initialised senders.

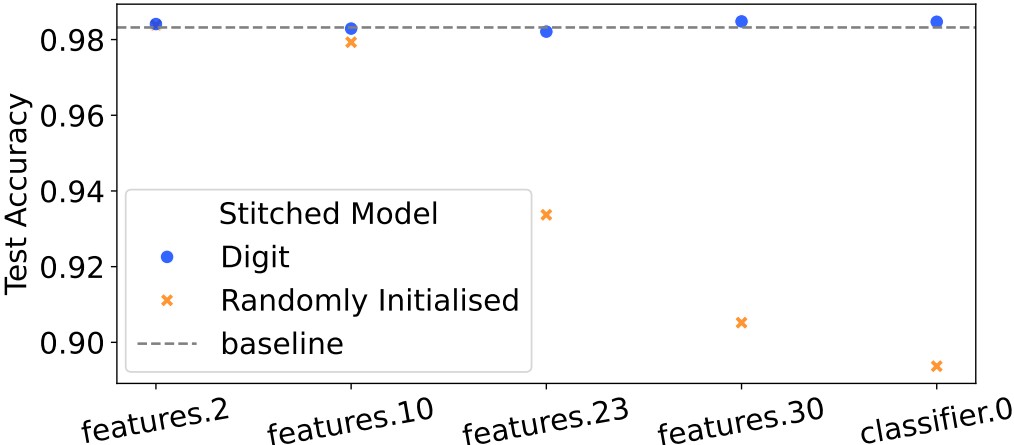

*Figure G.3.* VGG19 model (trained on Digit) stitched to itself (using Digit dataset) compared with stitching in an uninitialised network using the same dataset. Baseline is whole Digit model on Digit dataset.

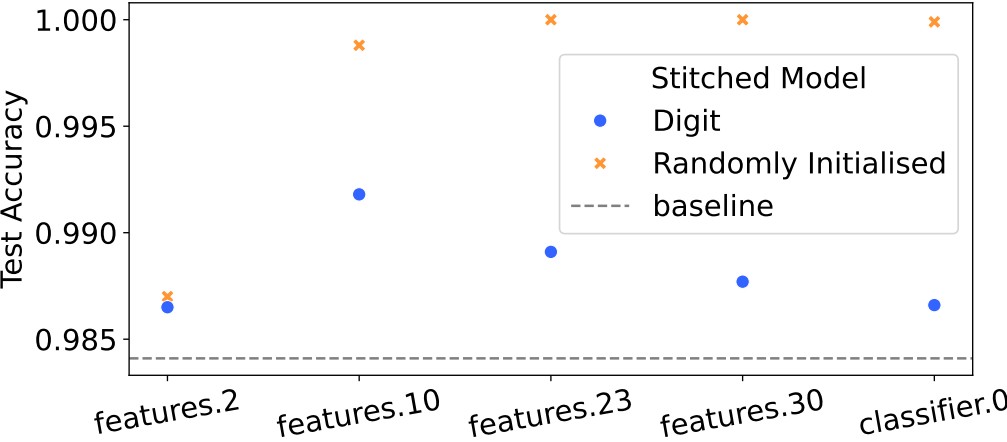

*Figure G.4.* VGG19 model (trained on Digit) stitched to itself (using Correlated dataset) compared with stitching in an uninitialised network using the same dataset. Baseline is whole Digit model tested with Correlated dataset.

|  | Conv 1 | Avg Pool 1 | Conv2 | Avg Pool 2 |
|---|---|---|---|---|
| Colour-Only | $95.33_{\pm 6.00}$ | $98.88_{\pm 0.76}$ | $96.94_{\pm 2.68}$ | $88.20_{\pm 9.07}$ |
| Greyscale MNIST | $95.12_{\pm 0.85}$ | $92.78_{\pm 0.42}$ | $90.04_{\pm 0.43}$ | $86.15_{\pm 0.91}$ |

*Table G.1.* Test accuracy when probing untrained LeNet-like senders for Colour-Only and Grayscale MNIST data.

|  | Block 1 | Block 2 | Block 3 | Block 4 | Block 5 | Linear |
|---|---|---|---|---|---|---|
| Colour-Only | $100_{\pm 0.00}$ | $100_{\pm 0.00}$ | $100_{\pm 0.00}$ | $100_{\pm 0.00}$ | $100_{\pm 0.00}$ | $100_{\pm 0.00}$ |
| Greyscale MNIST | $97.94_{\pm 0.07}$ | $98.33_{\pm 0.09}$ | $98.26_{\pm 0.11}$ | $97.01_{\pm 0.21}$ | $93.77_{\pm 0.40}$ | $82.88_{\pm 0.49}$ |

*Table G.2.* Test accuracy when probing untrained ResNet-18 senders for Colour-Only and Grayscale MNIST data.

To do so, we linearly probe these representations. That is, we put the representations through a linear layer and attempt to classify these. We train the probe (linear layer) for 20 epochs, using SGD with a momentum of 0.9 and weight decay of $10^{-4}$, learning rate of $10^{-3}$, batch size of 128. We perform 5 runs and report the mean and standard deviation.

Note that we did not perform any hyperparameter tuning and it is possible that training for longer or with more carefully chosen hyperparameters, a higher degree of linear separability might be found. We simply want to show that good separability can be found. We perform these experiments for the Colour-Only and Greyscale MNIST datasets. The results for LeNet-like untrained senders are given in Table G.1, while those for untrained ResNet-18 senders can be found in Table G.2.

### G.5. Discussion on Randomly Initialised Senders

In this section we have seen that there are cases in which untrained versions of the models we considered *can* be successfully stitched (and also cases in which this was not achieved with the standard hyperparameters and training budget.) Are our results diminished by cases where stitching from an untrained model was possible? We believe they further amplify the central message of our paper.

Consider the case of putting Colour-Only data through untrained convolutions. As we have seen, there are many cases where colour information leaks through untrained convolutions, and representations remain clustered (see Appendix G.4 where we verify this in a couple of cases with a linear probe). In the light of the experiments in Section 5, we argue that stitch compatibility of untrained senders can be understood from the perspective of clustering. As long as a mapping can be found, we can abstract away from the information that the representational clusters were originally based on. Whether it is birdsongs, natural images, stylised images, random clustered noise, or even colour patches passed through untrained convolutions, the stitch need only find a mapping to the receiver. The stitch and receiver become a complex but fixed probe.

