# OpenReview forum: "Functional Alignment Can Mislead: Examining Model Stitching"
_ICML.cc/2025/Conference — ICML 2025 spotlightposter_

### Official Review · Reviewer_rqks · 2025-03-02

**Overall Recommendation:** 1

**Summary:**

The authors investigate the suitability of model stitching as a tool for analyzing the informational content of feature representations. Specifically, the authors show that models trained for different tasks (which arguably encode different information in their representations) could be stitched together to achieve high accuracy on a corresponding dataset. The authors demonstrate this behavior in several settings: in a controlled experiment using the Colored MNIST task, in a more realistic setting using ImageNet-type datasets, and in an autoencoding task. These results indicate that the model stitching does not identify crucial differences between feature representations, which limits its applicability for representational comparison.

## update after rebuttal

I am inclined to keep my current score. As I argued in my Rebuttal Comment, I think the current methodology generally does not test the model stitching as the measure of the informational content of the **initial representation function**, and instead tests the similarity of the **final classification functions**. Thus, I still think that the current version is **misleading**.

Specifically, the experiments in Section 4 indeed test the inapplicability of model stitching as the measure of **representation function** similarity. However, the message of this section is weaker than the message of the paper: it only identifies the **failure mode** of model stitching in the situation **where the stitching dataset has two distinct patterns that perfectly explain the data labelling**.

As for Section 5, I still insist that these experiments are different in scope and test only the applicability of model stitching for the comparison of **final classification functions**, and not the **initial representation functions**. I think this result is weaker and should be positioned differently.

**Claims And Evidence:**

I have some problems with comprehending the paper's explanations. So, I start with an outline of my understanding of the paper's methodology to clarify my evaluation.
1. As I understand, when model $A$ trained on dataset $D_A$ (sender) is stitched to model $B$ trained on dataset $D_B$ (receiver), we get a new model $C$ that starts (from input) with model $A$'s layers and ends with model $B$'s layers.
2. By default, the stitching layer of model $C$ is trained on the training portion of dataset $D_B$.
3. By default, model $C$ is evaluated on the test portion of dataset $D_B$.
4. By stitching "clustered noise" to models, the authors mean stitching models trained on clustered noise to models.
5. All stitches with models trained on clustered noise were evaluated on the receiver's dataset.

Now, I describe my evaluation of the paper's claims.
1. Given my understanding, the results in Table 1 can not be correct. Specifically, in the last row, the stitched models reach 100% accuracy on ImageNet, which is impossible for ResNet-50. Additionally, it casts doubt on the results of Section 4.2 for the random noise stitched with Digit models. If I have misunderstood and the stitched network was evaluated on the noise dataset, I have two comments. First, the paper should clearly mention it in the main text. Second, these results do not provide much insight because the remaining receiver layers were trained for a classification with extracted features, which means stitching a trained feature extractor to them would be easy.
2. A few experiments in the paper fail to distinguish between two possible explanations for the results: explanation via the properties of the similarity metric and explanation via the properties of a learning task. For example, it could indeed be possible that the representations of models trained on Stylized ImageNet are "worse" for usual ImageNet than those of models trained on usual ImageNet because the models trained on usual ImageNet capture additional texture information. If this is the case, model stitching indeed gave us valuable information about the learning task itself.
3. The results in Section 4 lack a discussion of crucial reference points for comparison. Bansal et al. (2021) introduced two reference points: the performance of the receiver model and the performance of the randomly initialized model stitched with the receiver. In contrast, the current paper only analyzes the performance of the receiver model and does not discuss the performance of the randomly initialized model stitched with the receiver. The second reference point is crucial for the interpretation of the results. For example, if the stitch of a randomly initialized model achieves high accuracy, it means that the stitch is too powerful for the MNIST task, strengthening the concern expressed at the end of Section 4.
4. The previous point also suggests that the paper should discuss the choice of stitching family for comparisons. For example, 1x1 convolutions might be indeed too powerful for meaningful comparison of representations for MNIST data. However, simple scalar rescaling of channels or strongly regularized 1x1 convolutions might still provide useful information about representations.
5. Given that I have concerns about the experiments on MNIST data and stitches of random clustered noise with ImageNet receiver, I do not think the paper definitively shows that model stitching does not provide useful information about representation similarity.

**Essential References Not Discussed:**

All essential references seem to be discussed.

**Experimental Designs Or Analyses:**

I find some experiments not informative. I think the most informative designs were presented in Sections 4.2, 5.2, and 5.4. I would prefer the paper to either focus on these results or discuss other results' limitations more.

**Methods And Evaluation Criteria:**

I think the scope (ResNet-18 models on MNIST and ResNet-50 models on ImageNet) is sufficient. However, I think ResNet-18 model is too powerful for MNIST dataset. A better choice would be the LeNet-5 model or a simple convolutional model. Additionally, I think performing a hyper-parameter sweep to choose the stitching layer's learning rate would be more appropriate than using a fixed one.

**Other Comments Or Suggestions:**

At the beginning of Section 5.4, the phrase "to stitch a model trained to recognise bird songs to a pretrained ImageNet model" contradicts the discussion after and Table 1. It should be the opposite "to stitch a pretrained ImageNet model to a model trained to recognise bird songs".

**Other Strengths And Weaknesses:**

I think the paper has some issues with writing and clarity.
1. The discussion of noise at the end of Section 4.1 is confusing because the authors explain their construction of noise dataset only in Section 4.2.
2. It is often hard to understand which datasets were used for model training, stitch training, and stitched model evaluation. For instance, I did not understand the textual definition of the sender's baseline and receiver's baseline in Section 5, and only managed to understand these notions after looking at Table D.1.
3. I have trouble understanding the motivation and interpretation of some results. For example, I do not understand what we could infer from Section 6. This section does not directly study the considered model stitching metric, and basically only demonstrates qualitative results. At the same time, the finding that MNIST decoder will produce MNIST-like images does not seem surprising for me since this decoder can not produce anything else.

**Questions For Authors:**

1. Do I correctly understand your methodology for model stitching?
2. How did your stitch of random noise with ImageNet achieve 100% accuracy on the ImageNet dataset?
3. Could you elaborate (compared to Section 7) on the cases where the model stitching could provide valuable information?
4. Would you get different results if you used a less powerful stitch for the Colored MNIST dataset?
5. Would you get different results if you used a less powerful architecture for the Colored MNIST dataset?
6. How would you determine the quality of representations for a specific task without using the model stitching method?

**Relation To Broader Scientific Literature:**

The paper evaluates the limitations of a particular representation similarity metric. This investigation could be important for the deep learning theory and interpretability literature since representation similarity is an important concept in these fields.

**Theoretical Claims:**

There are no theoretical claims.

---

> ### Author Rebuttal · Authors · 2025-03-31
>
> We sincerely thank the reviewer for stating their understanding of the paper and for the time they dedicated to structure the evaluation of our paper.
>
> ## Questions For Authors
>
> * Q1: The reviewer has misunderstood our methodology. While point 1 is correct, points 2--5 are incorrect.
>     * 2&3: we are training and evaluating the stitch on dataset $D_{A}$.
>     * 4: We are not stitching a model that was **trained** on clustered noise to a receiver. We remove the sender and supply the stitch with clustered random noise matching the dimension of the expected feature maps.
>     * 5: Since there is no sender, the stitched model is evaluated on newly sampled random noise using the same cluster centres as in the training set.
> * Q2: 100% accuracy was achieved on clustered noise, not on the ImageNet data set.
> * Q3: Does the reviewer mean L417 “While there might be cases in which stitching is insightful”? All we wanted to say is that we don’t exclude the possibility that applications could be found. We did consider several possible interpretations of model stitching, but did not find stitching to be insightful. In L421-427 we discuss why we believe other methods are more suitable.
> * Q4: The reviewer suggested two alternatives for stitching: 1. Rescaling of channels: Due to network symmetries this would not reliably work even if two networks were capturing the same information. 2. Regularisation: to address this we experimented with L1 of varying strength and our claims remain valid. Please see our response to reviewer WvUr for details.
> * Q5: We have re-run some of our spurious correlation experiments with a LeNet implementation compatible with a 28x28 input size. Our findings remain valid for this network. For LeNet we also tried stitching to a **randomly initialised** sender as suggested by the reviewer and we can confirm that stitching fails in this case, further strengthening our results.
> * Q6: Our paper shows stitching's inability to identify dissimilarity between networks. It is outside the scope to propose **model quality** tools. Nonetheless, we agree that the community should aim to define additional notions of model quality beyond performance on held-out data, robustness, calibration, representation clustering, etc.
>
> ## Other Comments or Suggestions
> We thank the reviewer for pointing this ambiguity. We had stitched the ImageNet sender to a Birdsong receiver. We have now clarified this in the manuscript.
>
> ## Other Strengths and Weaknesses
>
> * W1: Thank you. We have now fixed this.
> * W2: Thank you. We created a diagram further explaining the methodology (including data sets used for training and evaluating the stitches) and will include it in the revised paper.
> * W3: A standard AE trained on MNIST doesn't reliably generate meaningful images, particularly from arbitrary latent space samples (a key VAE motivation). Please let us know if the interpretation of these results in L369–373 needs extending. The relevance to the community was noted by reviewers HH2d and 93zT.
>
> ## Supplementary Material
>
> The contradiction stems from the reviewer’s misunderstanding (Q1). Thank you for highlighting the missing bit of code. Upon publication the full repository will be publicly released.
>
> ## Methods and Evaluation Criteria
>
> For LeNet, see Q5. We did not tune hyperparameters as it suffices to show that one map between different representations exists. We believe showing that we can stitch even without tuning hyperparameters makes a more compelling case.
>
> ## Claims and Evidence
>
> * E1: See Q1.4 and Q2.
> * E2: This alternative explanation motivated our core experiments on artificial data sets. There, we know exactly what patterns exist in the data refuting the alternative. Experiments on real-world data complement them. In L253--260, we acknowledge that in the absence of ground truth, alternative interpretations exist.
> * E3: Note that Bansal et al. (2021) do not introduce the performance of the randomly initialised network as a precondition for the applicability of the model stitching to a specific context. Rather, they use it as evidence that their proposed stitching methodology works. Nonetheless, we have now performed this reference experiment on a few models. For the considered experiments we find that randomly initialised networks failed to stitch, yet our models were successfully stitched together. Therefore our observations hold even with this additional requirement (although it was not proposed as a requirement by Bansal et al. (2021)).
> * E4: See Q4 and E3.
> * E5: Could the reviewer let us know if they have any remaining concerns and how they would need these to be addressed? We would like to reiterate that the point we are trying to make is that while model stitching **may** correctly identify equivalent networks as **informationally equivalent**, we provided multiple counterexamples in which stitching incorrectly identifies as equivalent networks that process semantically distinct types of input data.

---

> > ### Comment · Reviewer_rqks · 2025-04-02
> >
> > Thanks for the response!
> >
> > The authors addressed all my questions. However, given the authors' clarification about their methodology, I became more concerned about the paper's results.
> >
> > The biggest problem is that **the paper currently does not test the applicability of model stitching** described in Bansal et al. (2021). Section 2 of Bansal et al. (2021) suggests training a stitch on loss $\mathcal{L}$ to merge a representation function $r$ (i.e., the initial layers of the sender) with the rest of the model $A_{> \ell}$ (i.e., the final layers of receiver). While the beginning of this section does not necessarily clarify the dataset used for training and evaluating a stitch, the sentence "In this case, model stitching tells us if there is a way to linearly transform the representation $r$ into that of the first layers of $A$, ..." clearly suggests that **the stitch is trained on the dataset associated with $A$ (i.e., the receiver's dataset, which was denoted in my review as $D_B$)**.
> >
> > Currently, the paper tests a procedure that could be called **reverse model stitching**, i.e., instead of testing the applicability of the representation function for a new task, the current procedure tests the applicability of the final "classification" function. First, I think this is very misleading for the readers since it is a non-standard definition. Second, I think it weakens the results. I am much less surprised that the classification layer, which was trained to receive processed features for classification, is well suited for a classification task with the same number of classes and another set of processed features. One possible explanation for this phenomenon is the information bottleneck principle (Shwartz-Ziv & Tishby, 2017), which suggests that the final layers of the network will not rely on information about the distribution of inputs.
> >
> > In any case, I think either the paper should be heavily rewritten, or the experiments should be done following the original methodology of Bansal et al. (2021).
> >
> > **References**
> >
> > Ravid Shwartz-Ziv, Naftali Tishby (2017). Opening the Black Box of Deep Neural Networks via Information. arXiv:1703.00810

---

> > > ### Author Response · Authors · 2025-04-02
> > >
> > > We thank the reviewer for their response. 1) It is important to address the question of [1]’s original setting. **All our core experiments (Sec 4) are on [1]'s original setting** as we will reiterate below; 2) We assert that the extension to the multi-dataset setting (Sec 5) that the reviewer is concerned about is fully justified in the light of the central message of the paper, supported by Sec 4; 3) We think that the reviewer has managed to highlight **exactly the problem that [1]’s model stitching faces**. That is, the reviewer’s criticism of the extended setting (Sec 5) is exactly our criticism of the original model stitching. Because networks compress representations, stitching cannot be expected to distinguish between models that capture different input patterns as long as they compress and separate the representations sufficiently.
> > >
> > > ## 1. All our **core** experiments are on [1]'s original setting
> > >
> > > [1] stitch together senders and receivers trained to solve the same task (i.e. achieve good test accuracy on the same dataset). This is exactly what we do in our core experiments (Sec 4.1). We assume we are given the same **task** for both the sender and the receiver -- the Correlated dataset. Both the sender and the receiver achieve high test accuracy on the Correlated dataset. **We stitch using the same dataset** - i.e. Correlated. Stitching incorrectly finds that models which learned different shortcuts represent the same information. In the case of our core experiments **the reviewer's concern does not stand since we are effectively using the same setting as [1]**. Please see L110--133 (rhs column). Therefore, the point raised by the reviewer is irrelevant to our core experiments and main claim.
> > >
> > > We will next justify why it is also irrelevant for the rest of the paper.
> > >
> > > ## 2. Extending the message of the core experiment - different datasets for sender and receiver
> > >
> > > In Sec 5, we provide broader illustrations of our argument by considering multiple datasets. **It is only at this point that the reviewer’s concern about datasets comes into discussion.** Note that [1] did not consider different datasets in the first place which may be why they did not propose a stitching methodology for this situation. We believe that because [1] are using the same dataset for sender, receiver, and stitching, their setting occludes an important issue: classification compatibility can be found between very different representations.
> > >
> > > The point of Sec 5 was to challenge the assumption of [1]: stitching is only possible if similar information is represented. We do this by illustrating the potency and agnosticism of the stitching process. Fundamentally, stitching takes representations and attempts to map them to a receiver for the purpose of classifying them. We demonstrate that this could successfully be taken to an extreme even where the information represented by the sender, and that of the first layers of the original receiver, is different (Sec 4.2 and 5). Because of this, **a successful stitching in the original setting (same dataset for the sender, receiver, and stitch) cannot be interpreted as indicating that equivalent information is represented**. Please see L144--152. **The “reversed stitching” makes a broader statement, beyond the original setting which we had already addressed in Sec 4.1.**
> > >
> > > ## 3. Impact of the classification task
> > >
> > > > I am much less surprised that the classification layer, which was trained to receive processed features for classification, is well suited for a classification task with the same number of classes and another set of processed features.
> > >
> > > This is exactly what **[1]’s original stitching** (i.e. on the same dataset) does, and therefore it should not be surprising that it cannot distinguish between representations that capture very different information. We agree that as long as the sender network produces representations that can be separated for the purpose of classification, the information captured by the sender and first part of the receiver becomes irrelevant. Therefore, stitching cannot reliably identify informational similarity.
> > >
> > > To conclude, given that our controlled experiments work directly with [1]'s setting (same task for sender and receiver) we do not think the reviewer's concern is justified. The later experiments take this argument further by considering a setting where the sender and receiver learned to represent different input patterns, achieved through training on different tasks. The motivation is to show that a classification task can still be solved (which is what stitching fundamentally tests for), as expected based on information bottleneck-inspired intuitions and contrary to the assumption behind stitching. Therefore, we believe the specific criticisms of the reviewer do not hold and do not invalidate in any way our methodology and findings.
> > >
> > > [1] Bansal, Y., Nakkiran, P., and Barak, B. Revisiting Model Stitching to Compare Neural Representations, June 2021.

---

### Official Review · Reviewer_93zT · 2025-03-03

**Overall Recommendation:** 4

**Summary:**

This work conducts an empirical evaluation of when functional alignment is not an indicator of the semantic similarity of the learned features. Functional alignment between models $A$ and $B$ is measured by the performance of a “stitched” model where an affine layer connects the first $l$ layers of $A$ and the last $l$ layers of $B$ (where $A$ is  the “sender” and $B$ is the “receiver”). Experiments in a controlled setting are conducted, where the stitched model is evaluated on a colored MNIST dataset for which both digit and background color correlate with the target label. When the sender and receiver models are trained to represent different semantic features, the stitched model achieves improved performance, demonstrating that functional alignment can be high in spite of the models learning different semantics. Additional results on autoencoders and real world datasets support the claim that model stitching performance is not necessarily indicative of information similarity.

## update after rebuttal
I will keep my current score, as my clarification questions have been addressed.

**Claims And Evidence:**

Yes.

**Essential References Not Discussed:**

None that I know of.

**Experimental Designs Or Analyses:**

Yes, I checked the stitching experiments and looked through the Appendix B. One potential concern is that no hyperparameter tuning was done for the sender or receiver models on the real-world data (although many experiments use pretrained ImageNet models).

**Methods And Evaluation Criteria:**

Yes.

**Other Comments Or Suggestions:**

N/A.

**Other Strengths And Weaknesses:**

Strengths
1. The problem of understanding whether model stitching is misleading is interesting and well-motivated.
2. Experiments in the controlled colored MNIST setting support the claim that high stitching performance can be achieved without models representing similar information.
3. Experiments outside of the colored MNIST setting are comprehensive, as the authors experiment with stitching cross-modal representations and autoencoders.

Weaknesses
1. It doesn’t seem like hyperparameters for the sender or receiver models were tuned, which may be a concern for real world data (particularly when the accuracy is not close to 100%).
2. In Section 7, the authors present a potential hypothesis that high functional alignment is indicative of the clustering quality of the sender’s representations. However this hypothesis is not fully explored in the sense that the number of clusters (classes) that the sender is trained on is the same as the number that the receiver is expecting. It would be interesting to see how the stitching performance varies when the number of clusters is misaligned.

**Questions For Authors:**

1. Would tuning hyperparameters in the autoencoder setting improve the performance of the class reconstruction method?
2. How does the stitching performance affected when the number of clusters that the sender and receiver models are trained to represent are not the same? For example, if the number of clusters in the noisy data is less than 10, would stitching always yield worse performance?

While I believe these experiments would strengthen the paper, I think that the evidence in the paper that functional alignment is misleading is compelling and well-supported.

**Relation To Broader Scientific Literature:**

Model stitching has been studied in the past [1, 2] as a way to measure the representation similarity between neural networks trained with different settings (e.g.  initializations, different datasets). These works find that better performing networks have higher functional alignment and assume that the reason for the high alignment is due to different representations capturing similar information. This paper probes whether this intuition holds by controlling for the information that the networks can learn through varying the training data.

[1] Bansal et al. “Revisiting Model Stitching to Compare Neural Representations” (2020).

[2] Csiszarik et al. “Similarity and Matching of Neural Network Representations” (2021).

**Theoretical Claims:**

N/A.

---

> ### Author Rebuttal · Authors · 2025-03-31
>
> Thank you for taking the time to review our paper and for the positive comments.
>
> ## Experimental Design or Analyses:
>
> > One potential concern is that no hyperparameter tuning was done for the sender or receiver models on the real-world data (although many experiments use pretrained ImageNet models).
>
> To address this, we started experimenting with a few different learning rates and momentum values. So far we have not noted any significant improvement in performance but we are happy to try additional configurations if the reviewer believes this would impact the quality of the paper. However, we believe that tuning the hyperparameters would neither increase nor decrease the support for the point we are trying to make. If model stitching is a reliable model comparison method, it should work for both high-performing models as well as models trained without hyperparameter tuning.
>
> ## Other Strengths and Weaknesses
>
> * W1: See above our answer to the Experimental Design and Analyses section.
> * W2: We ran the experiment suggested by the reviewer. We modified the noise data to represent only a small number of classes and mapped it to an ImageNet-trained receiver (1000 classes). We experimented with 10, 5, and 2 classes for the noise data. We achieved full stitching compatibility in all these cases. This is not surprising, since the stitch only needs to be able to map the representations of the 10 clusters to any subset of size 10 of the 1000 ImageNet classes. As long as this mapping can be learned, the remaining 990 classes can be ignored. Did we correctly understand the experiment suggested by the reviewer?
>
> ## Questions for Authors
>
> * Q1: We would like to thank the reviewer for pointing out this missing detail. For the autoencoder experiments had already tuned the learning rate, but omitted to state this in the manuscript. It is possible we may get better performance by further tuning the hyperparameters. However, the current setup is sufficient to demonstrate that stitching **can** align seemingly disparate representations in the unsupervised/generative setting. We are happy to consider further hyperparameter tuning if the reviewer believes this would significantly improve our paper, but we believe the same argument as in the discriminative case applies.
>
> * Q2: See W2.
>
> We would kindly ask the reviewer to let us know if there are any concerns we did not address to a satisfactory level.

---

> > ### Comment · Reviewer_93zT · 2025-04-03
> >
> > I thank the authors for the response and additional experiments. For the W2 experiment, the setup of having fewer clusters in the noisy data than the number of classes that the receiver model is trained on is indeed what I meant. But my question was more about whether a stitch that is trained and evaluated on the receiver dataset predictably degrades (for example, the performance of stitching 100 noisy to 1000 classes would be better than 10 noisy to 1000 classes)? For the hyperparameter search results, it would be great to include them in the final paper for completeness.
> >
> > In light of the concerns brought up by reviewer rqks, I reread Section 4. In my understanding Section 4.1 experiments on the same setting proposed by Bhansal et al [1] – specifically, stitching compatibility is evaluated on the Correlated dataset, which has the same input examples as the Colors and Digits datasets. As either color or digit color can be used to achieve good performance on the Correlated dataset, the authors then experiment on the “reverse stitching” setting to exclude the possibility that the Color mode leaks information of shape to the Digit model. This is the point where I have some confusion, in that I’m unsure why the authors chose to do “reverse stitching” setup of training and evaluating on the sender dataset. An alternative experiment would be to stitch representations from a sender model trained on only colors (without any shape features) to the Digit model and use the correlated dataset for training and evaluating the stitch. Could the authors elaborate on whether they believe this proposed experiment would change any findings (if not in the paper) as well as clarify the motivation for the reverse stitching experiments? It isn’t clear to me how reverse stitching eliminates the possibility of information leakage.
> >
> > [1] Bansal et al. “Revisiting Model Stitching to Compare Neural Representations” (2020).

---

> > > ### Author Response · Authors · 2025-04-03
> > >
> > > We thank the reviewer for their response and for engaging with the broader conversation around the submission.
> > >
> > > We are happy to include in the revised version of the paper the additional experiments and the discussion about the number of classes. The effect of the number of classes will depend on the data set but it is expected to increase with a lower number of classes (it is typically easier to separate 10 classes compared to 100). As with hyperparameter tuning, the results will neither increase nor decrease the support for the point we are trying to make, but we agree that it might be interesting to further analyse the impact of the number of classes. We thank the reviewer for the suggestion to include this.
> > >
> > > > In my understanding Section 4.1 experiments on the same setting proposed by Bhansal et al [1]
> > >
> > > That is correct. We thank the reviewer for confirming that they agree with our statement.
> > >
> > > > This is the point where I have some confusion, in that I’m unsure why the authors chose to do “reverse stitching” setup of training and evaluating on the sender dataset
> > >
> > > The reason why we did “reverse stitching” is because we wanted to completely rule out alternative interpretations of our results. Even if the sender was trained on colour information only, it could still leak digit information since the correlated data contains both colour and digit. This is a possibility especially when stitching at the early convolutional layers or in networks with residual connections.
> > >
> > > >  An alternative experiment would be to stitch representations from a sender model trained on only colors (without any shape features) to the Digit model and use the correlated dataset for training and evaluating the stitch (..) Could the authors elaborate on whether they believe this proposed experiment would change any findings (if not in the paper)
> > >
> > > We performed this exact experiment and successfully managed to stitch (See Figure C.1. (a), Purple series: Colour-Only). Therefore, this experiment does not change our findings. However, this experiment does not fully rule out the possibility of information leak (see above). As a result, we designed a stricter setting, where the digit information is absent from the stitching dataset and therefore it is impossible for it to be leaked. See Sec 4.2 L210-213. Reiterating, the Colour-Only sender was not exposed to shape data during training, and also the data fed into it during stitching contains only pure colour (no digit). The stitch can still be formed successfully.
> > >
> > > **Concluding, we already did the experiment proposed by the reviewer as part of our core experiments. This was insufficient to prove that models which represent different information can be stitched together. The “reverse stitching” was an extension to allow us to make an unambiguous claim.** We hope we managed to clarify the justification of “reverse stitching” as necessary evidence for our conclusion. We thank the reviewer for the opportunity to reiterate the validity and motivation for our experiments.

---

### Official Review · Reviewer_WvUr · 2025-03-12

**Overall Recommendation:** 4

**Summary:**

This paper suggests rethinking the use of model stitching as a representation comparison tool. Model stitching is a functional approach to comparing representations of two models. Essentially one glues the first $k$ layers of one model with the last $\ell$ layers of another model with 1-2 *stitching* layers. The stitching layers are finetuned for the task while the components of the original models remain frozen. The idea is that if one model can use another’s activations to solve a task (with minor translation from the stitiching layers), then they must have had similar representations. The paper’s main point is that model stitching can show similarity in many cases where we would not expect similar representations and thus we should be careful about how we interpret model stitching results. Examples in the paper include: (i) models that have learned very different shortcuts on a dataset (color vs shape bias) can be effectively stitched together, (ii) models that have been trained to identify random noise can be stitched to a model trained on image data, and (iii) models trained on distinct modalities can be stitched together (images vs spectrograms). Finally, the paper argues that these results suggest fundamental limitations of stitching rather than other possibilities (such as convergent representations).

**Claims And Evidence:**

Overall, the reviewer believes that most of the core claims related to stitching are well-supported. It is where the paper makes claims about the functional perspective more generally that issues arise. For instance:

- Line 069: “We argue that in this context, the functional perspective alone does not lead to a meaningful comparison.” While the paper does a very good job providing support for the idea that stitching (in its usual form), is not reliable in many settings (including spurious correlations to which this quote refers), it does not provide much evidence that the functional perspective does not lead to meaningful comparison. This is a far stronger statement and would require an argument that takes into account all possible functional approaches to representation comparison (including those that don’t exist yet).
- Line 090: This issue where results about stitching are used as evidence against the functional approach generally appears again in a statement about functional approaches and model compatibility.

The reviewer believes that paper can stand on its own without these broader statements and would be improved if they were either softened or removed.

**Essential References Not Discussed:**

They are not essential but there are analogous works that have been performed for other types of representation similarity or XAI methods. It would be good to include some of these.

**Experimental Designs Or Analyses:**

Overall, the reviewer found the experimental design and analysis to be thorough and robust. There were a few small issues that stood out.

**Numerical rank:** The reviewer felt that the use of numerical rank as an invariant of representations could have been better motivated. There are a number of different invariants that one can calculate for representations (e.g., various geometric, topological, information-theoretic statistics). Numerical rank is a reasonable choice but certainly not the only one and some argument should be provided for why this was chosen among others. For instance, it may be that high-rank and low-rank representations only differ in relatively meaningless dimensions? That is, taking a high-rank representation and replacing it with a low-rank approximation would not actually change the behavior of the model?

**Methods And Evaluation Criteria:**

The reviewer was satisfied with the evaluation. The paper describes experiments in a satisfactory number of settings, including: (i) models learning different shortcuts, (ii) models learning on different datasets, and (iii) models learning on different modalities.

A deeper study of stitching hyperparameters, including different types of stitching layers, would help understand where and how stitching is failing to capture meaningful signal.

**Other Comments Or Suggestions:**

- The images in Figure three (3rd and 4th rows) have very odd colors. Are they supposed to look psychedelic?

**Other Strengths And Weaknesses:**

**Strengths:**

- **Clarity:** Overall, the paper is clearly written. The paper makes its arguments directly and concisely. The reviewer found this easy to read.

**Weaknesses:**

- **The issue of spurious correlations:** The reviewer found the experiments on spurious correlations interesting, but also worried that they are somewhat overshadowed by subsequent experiments which are much more dramatic. Stitching not catching whether two models use different shortcuts for a given task pales in comparison to stitching not catching the difference between two models trained for completely different modalities/tasks.

**Questions For Authors:**

My questions are implicit in the comments above.

**Relation To Broader Scientific Literature:**

The paper does a good job summarizing past work on stitching but the reviewer believes that it would be helpful to also situate the work within the line of research exploring the limitations of representation comparison methods and XAI methods in general. For instance [1] or [2].

The reviewer believes that this work is a strong contribution to that tradition.

[1] Davari, MohammadReza, et al. "Reliability of cka as a similarity measure in deep learning." arXiv preprint arXiv:2210.16156 (2022).

[2] Ding, Frances, Jean-Stanislas Denain, and Jacob Steinhardt. "Grounding representation similarity through statistical testing." Advances in Neural Information Processing Systems 34 (2021): 1556-1568.

**Theoretical Claims:**

The paper did not make any theoretical claims.

---

> ### Author Rebuttal · Authors · 2025-03-31
>
> Thank you for the useful suggestions and detailed review.
>
> ## Claims and Evidence
>
> We agree that the paper can stand on its own without the broader statements about the functional perspective. We will restrict these to the discussion section. Would this fully address the concern?
>
> ## Methods and Evaluation
>
> To address the point about different types of stitching we considered training the stitch with various degrees of L1 and L2 regularisation. For all the experiments we reproduced with the added regularisation term, we found that with mild regularisation, we are still able to stitch successfully. For more aggressive regularisation, we fail to stitch. To verify that this is due to learnability issues, we aim to stitch the same model to itself. That is, we break the same model instance up into a sender and a receiver and we try to stitch back up the model to itself (we know that there exists at least one mapping in this case - identity). This led to a stitching failure, indicating that for aggressive regularisation there is an issue of learnability that stops models from being stitched together successfully. In all the models where a model was successfully stitched to itself, we were also able to stitch between networks that learned different information. Therefore, our observations remain valid for different stitching options.
>
> ## Experimental Design or Analyses
>
> The numerical rank of representations provides a crude estimation of their compression and was used in recent publications [e.g.  1, 2]. We agree with the reviewer that the rank does not take into account which of the feature maps meaningfully contribute to the model’s output (i.e. are salient). However, we believe the alternative statistics proposed in the review suffer from the same limitation. We are unaware of any computationally feasible techniques which are reliably capturing this for the types of models we consider.
>
> We also agree that alternative invariants could be computed. However, we would equally struggle to motivate any particular one, since each has its limitations. If the reviewer believes there is a particular one which the paper would benefit from including, we are happy to add it in the revised manuscript. Note that our objective was simply to include additional evidence that the sender representations are not entirely equivalent when processed by the receiver (even if they only differ in meaningless dimensions). However, this point is far outweighed by our subsequent experiments and therefore the rank experiments could have been omitted altogether without affecting the contributions of the paper.
>
> [1] Masarczyk, W., Ostaszewski, M., Imani, E., Pascanu, R., Miłoś, P. and Trzcinski, T., 2023. The tunnel effect: Building data representations in deep neural networks. Advances in Neural Information Processing Systems, 36, pp.76772-76805.
>
> [2] Feng, R., Zheng, K., Huang, Y., Zhao, D., Jordan, M. and Zha, Z.J., 2022. Rank diminishing in deep neural networks. Advances in Neural Information Processing Systems, 35, pp.33054-33065.
>
> ## Relation to Broader Scientific Literature
> Thank you for the suggestion to situate our work within the broader representation comparison literature beyond functional methods and XAI more broadly. We are happy to include this in the manuscript.
>
> ## Other Strengths and Weaknesses
>
> We agree that the real-world experiments could be found more convincing than the artificial ones. However, as noted by reviewer HH2d, these artificial experiments allowed us to incrementally construct our argument, starting with settings where we had full control over the data and therefore could rule out alternative interpretations of our results. We treat the experiments on real-world data as additional evidence to our core experiments on spurious correlations, since in the real-world case we do not have a ground truth, and alternative explanations could be imagined. As far as we understand, reviewer rqks pointed out exactly these possible alternative explanations. Therefore, without controlled, incremental experiments, we would not be able to reliably show stitching’s limitations. We also believe that it is in the contexts where similar data is used that researchers are most likely to expect (and want to test for) the representation of similar information in their networks: therefore the core experiments are most likely to be persuasive and relevant to the intended audience.
>
> ## Other comments and suggestions
>
> Did the reviewer mean Figure 1 instead of 3? If so, we will adjust the saturation.
>
> We once again thank the reviewer for their time and would kindly ask them to let us know if we did not address the raised concerns.

---

> > ### Comment · Reviewer_WvUr · 2025-04-08
> >
> > We would like to thank the authors for their clarification and also the other reviewers for the helpful discussion.
> >
> > Having reviewed the paper, the other reviews, and the rebuttals, our assessment is that this work is likely to provide value to the community. The main issue seems to be the conclusions that were drawn from the experiments. In particular, some may feel that stronger claims are made than were warranted by the actual results. This reviewer is of the opinion that the experiments that are presented in this work tell us something interesting and say something important about model stitching. Removing some of the broader statements about functional comparison are a good step towards refocusing the work on the experimental results. The reviewer would encourage the authors to review other claims and adjust accordingly.
> >
> > As for the point about numerical rank, it would probably be enough to note that there are many invariants that one could apply (each of which has its limitations) and that the paper chose to use numerical rank.
> >
> > The reviewer enjoyed reading this paper, thanks!

---

> > > ### Author Response · Authors · 2025-04-08
> > >
> > > We are grateful to the reviewer for taking the time to consider not just our own discussion, but also the points raised by other reviewers.
> > >
> > > As agreed with reviewer HH2d, we are happy to remove the broader claims, especially given the paper’s central claims still stand without these.
> > >
> > > We are also happy to include a note about the choice of numerical rank and the existence of alternatives.
> > >
> > > We are very pleased that the reviewer enjoyed reading the paper, and felt it added value to the community.

---

### Official Review · Reviewer_HH2d · 2025-03-14

**Overall Recommendation:** 5

**Summary:**

This paper contributes to the representation alignment field. The core message is that the functional similarity of representation spaces, measured by stitching performance, is not correlated with information content. This is shown empirically through well-controlled settings (where the variation factors are known) in classification and autoencoding tasks. A final discussion is provided on the implications of this finding for the representation alignment research.

**Claims And Evidence:**

The claims made in this paper are exceptionally well-supported by empirical evidence. The experiments are not only well-designed but also structured incrementally, progressively introducing additional variation factors to strengthen the validity of the findings. Moreover, the results are clearly presented and effectively communicated.

**Essential References Not Discussed:**

I think references are generally well chosen and discussed, with a couple of exceptions:
- Section 1, line 071. Why is Bansal et al. 2021 cited for model stitching and not the (later correctly referenced) Lenc & Vedaldi 2015? Did the authors intend to target the "good networks learn similar representations" aspect?
- A couple of relevant works on representation alignment methods with stitching applications are missing:
  - [a] Proposes reusable components for stitching purposes, a missing reference for the stitching ones.
  - [b] Analyzes the zero-shot compatibility (mostly through stitching) of networks under various variation factors.

[a] Towards Reusable Network Components by Learning Compatible Representations; Michael Gygli, Jasper Uijlings, Vittorio Ferrari; AAAI 2021;
[b] Relative representations enable zero-shot latent space communication; Luca Moschella, Valentino Maiorca, Marco Fumero, Antonio Norelli, Francesco Locatello, Emanuele Rodolà; ICLR 2023;

**Experimental Designs Or Analyses:**

Experiments are particularly in line with the existing literature and look very solid, with clear experimental settings and accompanying results.

Even after reading the corresponding section in the supplementary material, I had trouble understanding the "Embedding Mapping" procedure for AE stitching (Section 6). My understanding is that, in this case, the stitching layer is optimized to map a sample from model A's encoding space to model B's encoding space, drawing samples from the joint dataset and not from any model-specific one. Can the authors kindly confirm/clarify this?

**Methods And Evaluation Criteria:**

This paper has no proposed "methods"; it revolves around analyzing the relationship between functional similarity and "information content". Therefore, model stitching performance is the evaluation itself. It is applied in the standard stitching procedure, with a (much appreciated) change in the reported baseline, stronger than the one commonly used in the literature.

**Other Comments Or Suggestions:**

- Personally, I would give more prominence to the autoencoding experiments, as they are among the most interesting ones, and they are not directly mentioned in the abstract/conclusions. The classification setting can be intuitively explained by decision boundaries accommodating linearly transferred samples, but the autoencoding case presents a more complex and intriguing scenario.
- In section 6, Fumero et al. is cited as "extending the definition of model stitching to the generative case". Stitching in autoencoders was previously shown in **[b]** (Moschella et al., 2023)
- In section 6.6, Class Mapping paragraph: CIFAR-10 is mentioned abruptly;
- In "Should stitching then be used as a measure of model quality?" (Section 7), I would suggest adding a reference to **[c]** at the end of the first paragraph, as this work directly ties the linear separability of the representations to the representational capacity.

**[c]** Towards an Improved Understanding and Utilization of Maximum Manifold Capacity Representations; Rylan Schaeffer, Victor Lecomte, Dhruv Bhandarkar Pai, Andres Carranza, Berivan Isik, Alyssa Unell, Mikail Khona, Thomas Yerxa, Yann LeCun, SueYeon Chung, Andrey Gromov, Ravid Shwartz-Ziv, Sanmi Koyejo; ArXiv 2024;

**Other Strengths And Weaknesses:**

### **Weaknesses**
- The experiments are relatively small-scale regarding the number of classes, datasets, and models used. Scaling these aspects up would further solidify the findings and make them more persuasive. That said, this does not diminish the significance of the results.

- The final paragraph of Section 7 (“Are our results possible because models reached a shared understanding of reality?”) could be reworded, specifying "functionally aligned". The Relative Representations paper (Moschella et al.) and The Platonic Representation Hypothesis (Huh et al.) both suggest that the structure of representation spaces is shared across models. In Moschella et al., it is shown that point-wise distances are similar across spaces, and in Huh et al., this compatibility analysis is scaled up in terms of models considered, but again, a similar structural metric is used (Jaccard similarity between sample neighbors across spaces), not a functional one. I would argue that a high structural alignment can, in fact, be seen as a somewhat shared conceptual representation.

### **Strengths**

I commend the authors for the clarity and logical flow of the paper and the incremental complexity of the experiments, which strengthens the validity of the claims.

**Questions For Authors:**

N/A (already addressed in the other sections).

**Relation To Broader Scientific Literature:**

The paper is well situated in the representation alignment literature, challenging the common belief that functional similarity is a good proxy for information content. I would go as far as saying that this paper is a must-read for anyone working in the field of representation alignment, especially as its findings and discussions imply a reconsideration of current practices and encourage more careful framing of results in future studies.

**Theoretical Claims:**

N/A

---

> ### Author Rebuttal · Authors · 2025-03-31
>
> Thank you for the useful recommendations, questions, and very constructive review.
>
>
> ## Experimental Design or Analyses
>
> > Even after reading the corresponding section in the supplementary material, I had trouble understanding the "Embedding Mapping" procedure for AE stitching (Section 6). My understanding is that, in this case, the stitching layer is optimized to map a sample from model A's encoding space to model B's encoding space, drawing samples from the joint dataset and not from any model-specific one. Can the authors kindly confirm/clarify this?
>
> Thank you for pointing out the missing clarification here. We believe the stated understanding is correct. It can effectively be considered that we create a joint dataset where we pair up each training sample from dataset A (used to train encoder A) with another training sample from dataset B (used to train encoder B). Training samples are paired up using the encodings (i.e. in the embedding space).  Specifically, we solve the linear sum assignment problem in the AEs bottleneck to pair up samples from datasets A and B. Passing an image from dataset A through encoder A, the stitch is trained to map it to its corresponding sample from dataset B, passed through encoder B. Does this help clarify our approach?
>
> ## Essential References Not Discussed
> Thank you for pointing out the miscitation on L071 and for the suggested additional references, which we have now included.
>
> ## Weaknesses
> > The experiments are relatively small-scale regarding the number of classes, datasets, and models used. Scaling these aspects up would further solidify the findings and make them more persuasive. That said, this does not diminish the significance of the results.
>
> We agree that including more models and data sets would further strengthen the point that model stitching is not adequately capturing information similarity in an even broader variety of settings. We also agree with you that this does not diminish the significance of our results since our contention is that it is sufficient to show that model stitching can be deceived into declaring incorrect matches in many cases (spurious correlations, different data sets, different modalities, etc.). These cases involve both carefully constructed artificial data sets as well as a number of real-world data sets including ImageNet. Note that to address the rebuttals, we carried out a number of additional experiments. These include a LeNet-like architecture for a number of our core experiments (requested by reviewer rqks) or stitching between non-matching number of classes (reviewer 93zT). We are happy to continue considering additional settings. Is there one particular data set or architecture that would significantly strengthen our paper? We will do our best to include these, provided that they are within our computational budget.
>
> >The final paragraph of Section 7 (“Are our results possible because models reached a shared understanding of reality?”) could be reworded, specifying "functionally aligned"
>
> We are happy to clarify that it is functional alignment that this paper targets (which Huh et al. use as supporting evidence, although their proposed method is indeed structural). Just to confirm, do you mean L369--381?
>
>
> ## Other comments or suggestions:
>
> * Thank you for the suggestion to emphasise the autoencoding experiments. We will address this in the revised manuscript. Would adding the following sentence to the abstract suffice “(...) We then show that clustered random noise, and models trained to solve entirely different tasks on different data modalities, can be successfully stitched into MNIST or ImageNet-trained models. _Even autoencoders trained on different data sets can be connected to each other._ (...)”?
>
> * “CIFAR-10 is mentioned abruptly”: Thank you for pointing this out. We will change this explanation to refer to Fashion-MNIST and MNIST, in line with the previous paragraph. We also added the missing CIFAR-10 citation.
>
> We would like to thank you once again for the positive review and suggested improvements. If we omitted something in our response or if our clarification did not answer your questions, we would kindly ask the reviewer to let us know.

---

> > ### Comment · Reviewer_HH2d · 2025-04-04
> >
> > Thank you for your work on the rebuttal! I don't have any remaining doubts. I read through all the other reviews/discussions and am confident in keeping the original score.
> >
> > ---
> >
> > Regarding my comment on the final paragraph of Section 7, yes. Let me break it down for clarity:
> >
> > > Therefore, we believe our experiments cast a shadow on the
> > interpretation of **representational alignment**.
> >
> > > we believe our results show that one needs to look beyond **representational alignment** to support this
> > claim.
> >
> > In these two cases, I believe "functional alignment"/similarity should be targeted, not representation alignment as a whole.
> >
> > > we believe similar arguments can be constructed for other types of alignment.
> >
> > Since the argument is mostly speculative and all the empirical results in the paper are on stitching, I feel there should at least be some conjecture or sketch of how this argument might be extended. Otherwise, it reads as a hypothetical overclaim without support.
> >
> > > We urge the community to rethink the ... Models that converge to a shared statistical model of reality might be aligned, but aligned models do not necessarily have a shared understanding of reality.
> >
> > I fully agree with this point, but what is it directed at? If the Platonic Representation Hypothesis is the target, I think it doesn't apply since the measured similarity there is structural, not functional, as in this work. If it's a general statement, then it would be clearer to explicitly frame it as a takeaway of this work (and again link it to functional similarity, not to the general alignment).
> >
> > Overall, the core of that comment is that this paragraph feels particularly strong and overly broad in its message about representational alignment without grounding beyond the functional similarity/stitching setting.

---

> > > ### Author Response · Authors · 2025-04-07
> > >
> > > We appreciate that the reviewer clearly stated that they read all other comments and they are confident in keeping their score. We thank the reviewer for confirming that we have resolved their questions and for engaging with the wider discussion around the paper.
> > >
> > > We appreciate the detailed clarification, which will help us ensure we are fully addressing the reviewer’s comment. We agree that it would be beneficial to clarify that it is functional alignment we are referring to in the highlighted sentences. We will also remove the “hypothetical” claim, as suggested, and address the strong and broad statements that the reviewer identified. We thank the reviewer for pointing these out.
> > >
> > > Finally, we thank the reviewer very much for all their involvement with our paper and for the thoroughness of their review.

---

### Decision · Program_Chairs · 2025-05-01

**Decision:**

Accept (spotlight poster)

**Comment:**

This paper studies the approach of comparing model information content via model studying, using case studies to illustrate failure modes of this analysis. The majority of the reviewers agree that the paper offers a useful contribution, that helps to clearly illustrate the difficulty of interpreting stitching results. While one reviewer maintains some concerns about the methodology and the overall interpretation of the results — which I agree could be further clarified in the final version of the paper — on the whole the other reviewers and I agree that this is a valuable and clearly-presented contribution to the literature.